# LiLAW: Lightweight Learnable Adaptive Weighting to Meta-Learn Sample Difficulty and Improve Noisy Training

## Abstract

Training deep neural networks in the presence of noisy labels and data heterogeneity is a major challenge. We introduce Lightweight Learnable Adaptive Weighting (LiLAW), a novel method that dynamically adjusts the loss weight of each training sample based on its evolving difficulty level, categorized as easy, moderate, or hard. Using only three learnable parameters, LiLAW adaptively prioritizes informative samples throughout training by updating these weights using a single mini-batch gradient descent step on the validation set after each training mini-batch, without requiring excessive hyperparameter tuning or a clean validation set. Extensive experiments across multiple general and medical imaging datasets, noise levels and types, loss functions, and architectures with and without pretraining demonstrate that LiLAW consistently enhances performance, even in high-noise environments. It is effective without heavy reliance on data augmentation or advanced regularization, highlighting its practicality. It offers a computationally efficient solution to boost model generalization and robustness in any neural network training setup. [1]

## 1 Introduction

The increasing availability of very large labeled datasets has played a major role in advancing machine learning and computer vision in recent years. However, imaging datasets often contain samples with varying levels of quality, affecting the efficiency and effectiveness of model training. This issue is particularly significant in medical imaging datasets, which typically have smaller sample sizes, exhibit greater heterogeneity, and often require specialized expertise for accurate labeling. Ensuring that models make the best use of noisy, heterogeneous data remains a key challenge, as not all samples are equally informative or beneficial for model performance. At varying rates at different points during training, samples can benefit the model, hurt the model, or not affect the model much at all.

Several methods have been proposed to quantify the difficulty or importance of individual samples (Agarwal et al., 2022; Paul et al., 2021; Baldock et al., 2021; Maini et al., 2022; Siddiqui et al., 2022; Rabanser et al., 2022; Toneva et al., 2018; Pliushch et al., 2022; Xu et al., 2021; Kong et al., 2021; Dong, 2023; Jiang et al., 2020; Seedat et al., 2024b). Understanding which samples a model finds difficult to predict is essential for safe model deployment, sample selection for human-in-the-loop auditing, and gaining insights into model behavior (Agarwal et al., 2022). Knowing or estimating example difficulty can help separate misclassified, mislabeled, and rare examples (Maini et al., 2022), prune data (Paul et al., 2021), quantify uncertainty (Dong, 2023), improve generalization (Xu et al., 2021), increase convergence speed (Kong et al., 2021), enhance out-of-distribution detection (Agarwal et al., 2022), and provide insights into example memorization and forgetting (Toneva et al., 2018). While existing methods provide insights into individual data points and how models learn, they often do not adjust the training process in a dynamic, efficient manner. This may restrict potential improvements in model performance, robustness, and generalization.

Focusing on the most relevant data at different stages of training can help leverage datasets effectively. A common limitation in many existing approaches is the static or uniform treatment of all data points during training. Standard practice does not apply additional weights to samples or take sample

---

[1] Code in Supplementary Material.

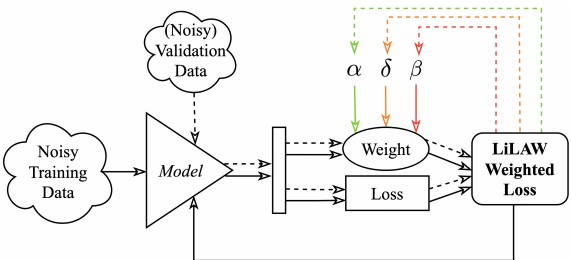

Figure 1: Given noisy training and validation data, LiLAW learns to adaptively weight the loss of each sample based on three trainable parameters, $\alpha, \delta, \beta$, pertaining to learning easy, moderate, and hard samples, respectively, at different stages of training using meta-learning on the validation data.

difficulty into account. Existing sample weighting methods usually require a clean validation set, a separate model to learn sample difficulty, and/or a higher training complexity (see Section 2.2).

In this paper, we propose a novel method called **Li**ghtweight **L**earnable **A**daptive **W**eighting (**LiLAW**), which addresses these challenges with a lightweight, adaptive mechanism to adjust the weight of each sample loss based on its difficulty. Our approach introduces a simple yet powerful parametrization, using three parameters pertaining to learning easy, moderate, and hard samples (see Figure 1). These parameters are used to calculate the weight of each sample loss and are jointly optimized during training with a single gradient descent step on the validation set after every training mini-batch.

Importantly, our method does not require the validation set to be clean and unbiased. Unlike static weighting schemes or hyperparameter-based weighting search schemes, LiLAW learns to adaptively prioritize different samples as training progresses. By integrating sample difficulty assessments into the training process, LiLAW helps the model focus on the most informative data when it is most beneficial. This approach offers a flexible solution for incorporating sample difficulty into any training process for classification problems, while maintaining computational efficiency.

We first introduce our theoretical framework and then perform extensive experiments with and without LiLAW in various settings. Our results show that LiLAW improves test performance across several general and medical imaging datasets, models, loss functions, noise types, noise levels, with and without pretraining, with and without calibration, and with and without a clean and unbiased validation set. This is done without the need for excessive hyperparameter tuning and without increasing the time and space complexity. In summary, we introduce a new method, LiLAW, that:

- only adds three extra trainable parameters and does not require more models for difficulty-learning or pruning (easy-to-implement and maintains similar space complexity);
- only needs one additional forward and backward pass on a single validation mini-batch after each training mini-batch (maintains similar time complexity);
- uses weighting to measure difficulty, inform model training, and improve test performance without pruning (avoids reducing dataset size, which may already be small in certain settings);
- improves noisy learning with label and input noise (prevalent in medical imaging datasets);
- uses meta-learning to avoid the need for extensive hyperparameter tuning (saves time);
- does not require a clean, unbiased meta-validation set (may be hard to obtain in health domain);
- offers an intuitive graphical and mathematical understanding of the weighting; and,
- is adaptive, lightweight, and easy-to-implement on top of any model training/fine-tuning.

## 2 RELATED WORK

The effectiveness of machine learning models, especially deep neural networks, heavily depends on the quality and proper use of training data. Recent literature has explored strategies to enhance model performance and robustness to address challenges brought forward by noisy, mislabeled, underrepresented, or otherwise hard-to-learn data.

## 2.1 Data-Centric AI

Pleiss et al. (2020) proposed using the area under the margin ranking to identify mislabeled data and improve model performance by filtering out mislabeled data, which requires "dataset cleaning." Toneva et al. (2018) conducted an empirical study on example forgetting during deep neural network training that shed light on data quality highlighting that some samples that are forgotten frequently, some are not forgotten at all, and some could be omitted from training without affecting the model greatly. Paul et al. (2021) highlight that simple scores such as the Gradient Normed (GraNd) and Error L2-Norm (EL2N) can be used to identify important examples very early in training and to prune significant portions of the training data while maintaining test accuracy. *Our method instead aims to improve test performance without pruning any training data.*

Wu et al. (2023) presented a framework for selecting pivotal samples for meta re-weighting, aiming to optimize performance on a small set of perfect samples, which may not be available in many real-world cases. Mindermann et al. (2022) note that a lot of samples may be hard to learn since they are noisy or unlearnable. They propose a method that selects points that are learnable, worth learning, and not yet learnt. However, this method requires a separate small model trained on a holdout set. *Our method does not require a clean and unbiased meta-validation set or a separate model.*

Agarwal et al. (2022) propose estimating example difficulty using variance of gradients to rank data by hardness and identify samples for auditing. Swayamdipta et al. (2020) present a method to map datasets using training dynamics to identify easy, hard, and ambiguous examples. These methods do not use difficulty information to inform training. *Our method serves both to assess sample difficulty and to use that information to enhance training.*

Jia et al. (2022) use raw training dynamics as input to an LSTM noise detector network which learns to predict mislabels. Jiang et al. (2021) used loss curves to identify corrupted labels and tailed classes by first training a whole network on the original data and then training a CurveNet on the loss curves of each sample as an additional attribute to identify bias type using meta-learning. These methods and others (Li et al., 2020; Wu et al., 2022; Karim et al., 2022; Liang et al., 2024) require additional models for difficulty-learning or data selection. *Our method does not require additional models.*

In addition, accurate uncertainty estimation and model calibration are essential for reliable predictions. Models are well-calibrated if the predicted probabilities accurately reflect the actual chance of an event occurring. Guo et al. (2017) investigated the calibration of neural networks and found that deeper networks are often poorly calibrated. They propose temperature scaling as an effective post-processing calibration method. Another approach to improve model calibration is label smoothing (Müller et al., 2020). *Our method works synergistically with calibration to improve model performance.*

Several methods were developed to characterize sample hardness and data quality such as Data-IQ (Data-Inherent Qualities) Seedat et al. (2022), DIPS (Data-centric Insights for Pseudo-labeling with Selection) Seedat et al. (2024a), and H-CAT (Hardness Characterization Analysis Toolkit) Seedat et al. (2024b). We use H-CAT to study the effectiveness of our method across various settings.

## 2.2 Sample Weighting

Sample weighting aims to improve model training by assigning weights to samples based on their learning difficulty. Traditional methods often rely on static weighting schemes, which fail to adapt to a model's evolving learning dynamics. Several works introduced a theoretical framework connecting generalization error to difficulty (Zhou & Wu, 2023; Zhou et al., 2023; Zhu et al., 2022).

Zhou & Wu (2023) propose adaptive weighting strategies that consider easy, moderate, and hard samples and utilize hyperparameter tuning to find the most effective weighting solution, which stays fixed, from easy-first, hard-first, medium-first, and two-ends-first or a pre-defined "varied modes" schedule (easy-first, then hard-first) during their training process.

Xu et al. (2021) mention that using counterfactual modeling to jointly train a weighting model with the classifier eventually causes the weights to converge to the same constant. Other methods state that increasing weights on hard samples may improve both convergence and performance, but assume that training noise is absent (Zhou et al., 2023; Xu et al., 2021). *Our method assumes that training noise (label noise or input noise) may exist and does not require extensive hyperparameter tuning to find the best learning strategy.*

Meta-learning aims to improve the learning process itself – it is often referred to as "learning to learn" in literature. Recent work has shown promise in using meta-learning to learn weights for samples. To aid in learning to select the samples for the meta-set, Jain et al. (2024) employ a bi-level objective aimed at identifying the most challenging samples in the training data to use as a validation set and subsequently train the classifier to reduce errors on those specific samples, which is a way of learning a Learned Reweighting (LRW) classifier without a fixed, unbiased, and clean validation set (Ren et al., 2019). However, this requires three networks: one to learn the task, one that identifies challenging samples to validate, and one that learns sample weights, vastly increasing computational complexity. *Our method only requires three additional trainable global parameters.*

Methods like probabilistic margin (Wang et al., 2022) and area under the margin (Pleiss et al., 2020) measure the difference between the score of the observed label in the softmax of the logits of the neural network and the maximum score in the softmax (discounting the score of the observed label). The vast range of possible negative values makes it difficult to use these methods effectively in a loss-based weighting scheme. *In contrast, our method guarantees reasonable positive weights.*

Kong et al. (2021) propose adaptive curriculum learning (as opposed to fixed) which uses the current model to adjust loss-based difficulty scores while retaining learned knowledge from a pre-trained model using the KL divergence between the outputs of the current model and the pre-trained model and a pacing function to control the learning pace. This assumes the existence of a pre-trained network for the task at hand. *Our method does not require additional models and only needs one additional forward and backward pass to do a single gradient descent step on the validation set after every training mini-batch.*

## 3 METHOD

Consider the following supervised multi-class classification problem setup used in prior work (Northcutt et al., 2022). Let $\mathcal{D}_t = \{(x_i, \widetilde{y_i})\}_{i=1}^N$ represent the training set and $\mathcal{D}_v = \{(x_j, \widetilde{y_j})\}_{j=N+1}^{N+M}$ represent the validation set. Note that $(x_i, \widetilde{y_i})$ represents the pairs of inputs and observed (potentially noisy) targets. Specifically, $x_i \in \mathcal{X}$, where $\mathcal{X}$ is the input space (e.g.: images) and $\widetilde{y_i} \in \mathcal{Y} = \{0, ..., c-1\}$, is the output space with $c \in \mathbb{N}$ such that $c \geq 2$ is the total number of classes. Note that $\widetilde{y_i}$ is a single integer value, i.e. $x_i$ belongs to a single class. Let $y_i \in \mathcal{Y}$ be the true target. Let $f_\theta : \mathcal{X} \to \mathcal{Y}$ be the neural network model, $\theta$ be its parameters, and $s_i = \texttt{softmax}(f_\theta(x_i))$ be the softmax of its logits. Therefore, $s_i \in \mathbb{R}^c$ such that $\sum_{j=0}^{c-1} s_i[j] = 1$. Note that $s_i[j]$, where $j \in \{0, ..., c-1\}$ refers to the softmaxed logit for class $j$ after passing input $x_i$ through the model.

Our method's motivation stems from the properties of two key values, $s_i[\widetilde{y_i}]$ and $\max(s_i)$:

– $s_i[\widetilde{y_i}] < \max(s_i)$ implies an incorrect prediction, if $\widetilde{y_i} = y_i$ (inconclusive otherwise)
– $s_i[\widetilde{y_i}] = \max(s_i)$ implies a correct prediction, if $\widetilde{y_i} = y_i$ (inconclusive otherwise)
– low $\max(s_i)$ implies an unconfident prediction
– high $\max(s_i)$ implies a confident prediction

Note that in the noisy setting, i.e. when the observed target is not the same as the true target (i.e. $\widetilde{y_i} \neq y_i$), we rely solely on the confidence of the prediction to inform sample difficulty. It is therefore possible for a sample to be predicted correctly and confidently, incorrectly and confidently, correctly and unconfidently, and incorrectly and unconfidently. The goal of LiLAW is to make unconfident predictions confident and incorrect predictions correct based on the two values mentioned above. See A.1 for a motivating example.

We have the following relations between $s_i[\widetilde{y_i}]$ and $\max(s_i)$ which define the area in Figure 2:

    i. $0 \leq s_i[\widetilde{y_i}] \leq \max(s_i)$,
    ii. $0 \leq s_i[\widetilde{y_i}] \leq 1$,
    iii. $0 < \max(s_i) \leq 1$,
    iv. Since $\sum_{j=0}^{c-1} s_i[j] = 1$, we know $s_i[\widetilde{y_i}] + \max(s_i) \leq 1$ when $s_i[\widetilde{y_i}] \neq \max(s_i)$.

The above constraints form a region, shown in gray in Figure 2, which contains the values that $s_i[\widetilde{y_i}]$ and $\max(s_i)$ can attain. The darker shading corresponds to areas of high loss and the lighter shading

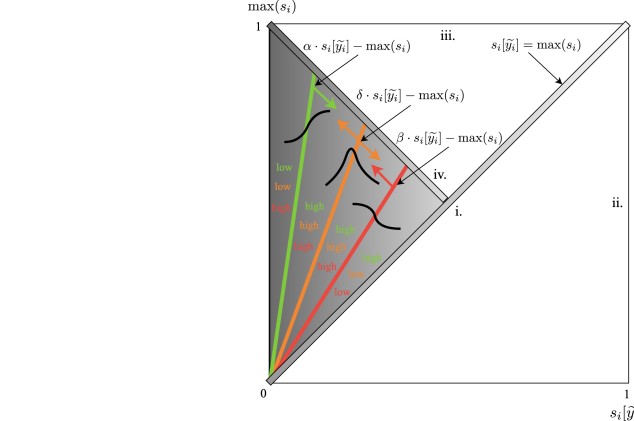

Figure 2: A graphical representation of the LiLAW weighting method. Darker areas correspond to high loss regions (due to incorrectness and/or unconfidence) and lighter areas correspond to low loss regions (due to correctness and/or confidence). The three weight functions in Equations (2), (3), and (4) correspond to the green, red, and orange lines, respectively, with the colored arrows representing the direction of descent for each corresponding weight function. Note also that Equations (2) and (3) use a sigmoid and Equation (4) uses a radial basis function. For the above configuration, we also signify whether each weight is high or low in a given region.

corresponds to lower loss. Note that the diagonal region going from the bottom left to the top right of the region corresponds to when $s_i[\widetilde{y_i}] = \max(s_i)$. Closer to the bottom left, we make unconfident predictions and closer to the top right, we make more confident predictions. The remaining shaded areas correspond to when $s_i[\widetilde{y_i}] < \max(s_i)$. Near the top left of the region, we have more confident predictions and near the bottom left, we have more unconfident predictions. Closer to the center of the region, we have somewhat unconfident predictions. Our method attempts to push unconfident, incorrect predictions towards being confident, correct predictions. As a result, it is able to deal with both label and input noise.

These properties provide granularity to help us identify predictions as being incorrect or correct (with respect to the observed label) and unconfident or confident. However, they are not sufficient to learn to adaptively weight the losses of certain samples throughout the course of training. Therefore, we use parameters to control the relationship between $s_i[\widetilde{y_i}]$ and $\max(s_i)$ during training. We define three scalar parameters, $\alpha, \beta, \delta$ to define the contribution of different data points to the loss function.

We define our regular (unweighted) loss function as follows (in our case, $\mathcal{L}$ is cross-entropy loss or focal loss (Lin et al., 2017)):

$$\mathcal{L} = \ell(f_\theta(x_i), \widetilde{y_i}) \tag{1}$$

We define the weights pertaining to $\alpha, \beta, \delta$ as follows:

$$\mathcal{W}_\alpha(s_i, \widetilde{y_i}) = \sigma(\alpha \cdot s_i[\widetilde{y_i}] - \max(s_i)) \tag{2}$$

$$\mathcal{W}_\beta(s_i, \widetilde{y_i}) = \sigma(-(\beta \cdot s_i[\widetilde{y_i}] - \max(s_i))) \tag{3}$$

$$\mathcal{W}_\delta(s_i, \widetilde{y_i}) = \exp\left(-\frac{(\delta \cdot s_i[\widetilde{y_i}] - \max(s_i))^2}{2}\right) \tag{4}$$

We calculated the weight for each sample as follows:

$$\mathcal{W}(s_i, \widetilde{y_i}) = \mathcal{W}_\alpha(s_i, \widetilde{y_i}) + \mathcal{W}_\beta(s_i, \widetilde{y_i}) + \mathcal{W}_\delta(s_i, \widetilde{y_i}) \tag{5}$$

and define our LiLAW (weighted) loss function as follows:

$$\mathcal{L}_W = \mathcal{W}(s_i, \widetilde{y_i}) \cdot \mathcal{L} \tag{6}$$

We define a sample's weight as the sum of $\mathcal{W}_\alpha$, $\mathcal{W}_\beta$, and $\mathcal{W}_\delta$ so that the final contribution of a sample is not governed by one parameter, but by the combined effect of all three. Easy, moderate, and hard samples primarily activate $\mathcal{W}_\alpha$, $\mathcal{W}_\beta$, and $\mathcal{W}_\delta$, respectively, while their overlap provides smooth transitions near boundaries and keeps LiLAW differentiable with respect to $\alpha, \beta, \delta$. Geometrically, $\mathcal{W}_\alpha$ is high when $\alpha$ is large and/or $s_i[\widetilde{y_i}] \ll \max(s_i)$, $\mathcal{W}_\beta$ when $\beta$ is small and/or $s_i[\widetilde{y_i}] \approx \max(s_i)$, and $\mathcal{W}_\delta$ when $\delta$ is large and/or $s_i[\widetilde{y_i}]$ is moderately close to $\max(s_i)$. We initialize $\alpha \leq \delta \leq \beta$, enforce $\alpha \geq 1$ to weight unconfident samples sufficiently, and require $\delta \geq \beta$ so that moderate samples receive more weight than easy ones but less than hard ones.

Now, let's consider $\mathcal{W}_\alpha = \sigma(\alpha \cdot s_i[\widetilde{y_i}] - \max(s_i))$.

$\mathcal{W}_\alpha < 0.5$ when:

$$\alpha \cdot s_i[\widetilde{y_i}] - \max(s_i) < 0 \implies \alpha \cdot s_i[\widetilde{y_i}] < \max(s_i) \leq 1 \text{ by (b)} \tag{7}$$

$\mathcal{W}_\alpha \geq 0.5$ when:

$$\alpha \cdot s_i[\widetilde{y_i}] - \max(s_i) \geq 0 \implies \alpha \cdot s_i[\widetilde{y_i}] \geq \max(s_i) \geq s_i[\widetilde{y_i}] \text{ by (a)} \tag{8}$$

As such, the upper and lower bounds on $\mathcal{W}_\alpha$, as seen in Figure 2, are as follows:

$$-1 \leq \alpha \cdot s_i[\widetilde{y_i}] - \max(s_i) \leq \alpha - 1 \implies 0.2689 \approx \sigma(-1) \leq \mathcal{W}_\alpha \leq \sigma(\alpha - 1) < 1 \tag{9}$$

Now, let's consider $\mathcal{W}_\beta = \sigma(-(\beta \cdot s_i[\widetilde{y_i}] - \max(s_i)))$.

$\mathcal{W}_\beta < 0.5$ when:

$$\beta \cdot s_i[\widetilde{y_i}] - \max(s_i) > 0 \implies \beta \cdot s_i[\widetilde{y_i}] > \max(s_i) \geq s_i[\widetilde{y_i}] \text{ by (a)} \tag{10}$$

$\mathcal{W}_\beta \geq 0.5$ when:

$$\beta \cdot s_i[\widetilde{y_i}] - \max(s_i) \leq 0 \implies \beta \cdot s_i[\widetilde{y_i}] \leq \max(s_i) \leq s_i[\widetilde{y_i}] \text{ by (b)} \tag{11}$$

The upper and lower bounds on $\mathcal{W}_\beta$, as seen in Figure 2, are as follows:

$$1 - \beta \leq \beta \cdot s_i[\widetilde{y_i}] - \max(s_i) \leq 1 \implies 0 < \sigma(1 - \beta) \leq \mathcal{W}_\beta \leq \sigma(1) \approx 0.7311 \tag{12}$$

Finally, consider $\mathcal{W}_\delta(s_i, \widetilde{y_i}) = \exp\left(-\frac{(\delta \cdot s_i[\widetilde{y_i}] - \max(s_i))^2}{2}\right)$. $\mathcal{W}_\delta = 1$ when:

$$\delta \cdot s_i[\widetilde{y_i}] - \max(s_i) = 0 \implies \delta \cdot s_i[\widetilde{y_i}] = \max(s_i) \tag{13}$$

Otherwise, $0 < \mathcal{W}_\delta < 1$, with $\mathcal{W}_\delta$ tending towards zero as $\delta \cdot s_i[\widetilde{y_i}]$ and $\max(s_i)$ grow apart.

During training (see Algorithm 1), we aim to find the parameters $\theta^*$ for the classifier that best minimize LiLAW loss on the training set using the three parameters $\alpha^*, \beta^*, \delta^*$ that best minimize LiLAW loss on the validation set to get the following objective:

$$\theta^* = \arg\min_\theta \sum_{(x_i, \widetilde{y_i}) \in \mathcal{D}_t} \mathcal{L}_{W_{\alpha^*, \beta^*, \delta^*}} \text{ such that } \alpha^*, \beta^*, \delta^* = \arg\min_{\alpha, \beta, \delta} \sum_{(x_i, \widetilde{y_i}) \in \mathcal{D}_v} \mathcal{L}_{W_{\theta^*}} \tag{14}$$

Note that $\mathcal{L}_{W_{\alpha^*, \beta^*, \delta^*}}$ refers to the loss being computed using the parameters $\alpha^*, \beta^*, \delta^*$ and $\mathcal{L}_{W_{\theta^*}}$ refers to the loss being computed using network parameters $\theta^*$. This objective is similar to the formulation in Jain et al. (2024); however, we do not require an instance-wise weighting function, but a more general weighting function based on $\alpha, \beta, \delta$. Similar to MOLERE (Jain et al., 2024), LRW (Ren et al., 2019), and MAML (Finn et al., 2017), we perform stochastic gradient updates on validation mini-batches to update the three parameters.

When it comes to $\mathcal{W}_\alpha$, $\mathcal{W}_\beta$, and $\mathcal{W}_\delta$, based on their corresponding definitions, we generally see that:

- $\mathcal{W}_\alpha$ is high when $\alpha$ is large and/or when $s_i[\widetilde{y_i}] \approx \max(s_i)$, i.e. when we have correct, confident predictions (*easy samples*)
- $\mathcal{W}_\beta$ is high when $\beta$ is small and/or when $s_i[\widetilde{y_i}] << \max(s_i)$, i.e. when we have incorrect, confident predictions (*hard samples*)
- $\mathcal{W}_\delta$ is high when $\delta$ is large and/or when $s_i[\widetilde{y_i}]$ is moderately close to $\max(s_i)$, i.e. when we have unconfident predictions (*moderate samples*)

---

**Algorithm 1** Training with LiLAW

---

1: **Inputs:** training set $(\mathcal{D}_t)$, validation set $(\mathcal{D}_v)$, classifier model $f_\theta$, LiLAW parameters $\alpha, \beta, \delta$,
2:          optimizer for $\alpha, \beta, \delta, \theta$, learning rates $\alpha_{lr}, \beta_{lr}, \delta_{lr}$, weight decay values $\alpha_{wd}, \beta_{wd}, \delta_{wd}$
3: **Output:** classifier model trained using LiLAW
4: **for** epoch $\leftarrow 1$ **to** $n$ **do**
5:    **for** batch $(x_t, \widetilde{y}_t) \leftarrow$ dataloader$(\mathcal{D}_t)$ **do**
6:      set requires_grad to False on $\alpha, \beta, \delta$, True on $\theta$
7:      $s = \texttt{softmax}(f_\theta(x_t))$
8:      calculate train loss $\mathcal{L}_W = \mathcal{W}(s, \widetilde{y}_t) \cdot \ell(f_\theta(x_t), \widetilde{y}_t)$
9:      $\mathcal{L}_W$.backward()
10:     optimizer.step() to update $\theta$
11:     optimizer.zero_grad()
12:     choose a random batch $(x_v, \widetilde{y}_v) \leftarrow$ dataloader$(\mathcal{D}_v)$
13:     set requires_grad to True on $\alpha, \beta, \delta$, False on $\theta$
14:     $s = \texttt{softmax}(f_\theta(x_v))$
15:     calculate val loss $\mathcal{L}_W = \mathcal{W}(s, \widetilde{y}_v) \cdot \ell(f_\theta(x_v), \widetilde{y}_v)$
16:     $\mathcal{L}_W$.backward()
17:     $\alpha = \alpha - \alpha_{lr} * (\nabla_\alpha \mathcal{L}_W + \alpha_{wd} * \alpha)$
18:     $\beta = \beta - \beta_{lr} * (\nabla_\beta \mathcal{L}_W + \beta_{wd} * \beta)$
19:     $\delta = \delta - \delta_{lr} * (\nabla_\delta \mathcal{L}_W + \delta_{wd} * \delta)$
20:     $\nabla_\alpha \mathcal{L}_W = \nabla_\beta \mathcal{L}_W = \nabla_\delta \mathcal{L}_W = 0$
21: **return** $f_\theta(x)$

---

Prior meta-reweighting work (Ren et al., 2019) assumes a small clean validation set to steer training toward the clean-label objective. Our setting is different and, in our opinion, more realistic: we explicitly allow the meta-validation set to contain label noise, because a clean set is often unavailable in practice. We do not claim convergence to the clean-label optimum, but adjust the three global parameters $(\alpha, \beta, \delta)$ so that the next training step improves performance on data drawn from the same source. It is still reasonable to use a noisy meta-validation set for LiLAW since: the meta-gradients for $(\alpha, \beta, \delta)$ are driven by confidence and agreement and the LiLAW weights are smooth and bounded, so no single meta example can dominate the update, which keeps the meta-learning dynamics stable.

## 4 EXPERIMENTS

We conducted extensive experiments with and without LiLAW to comprehensively assess its ability to boost test performance. We used H-CAT (Seedat et al., 2024b), an API interface for several hardness and data characterization techniques, to study LiLAW. We emphasize that our goal is not to achieve state-of-the-art results, but to demonstrate the effectiveness of LiLAW in various settings.

We also show how the weight functions (2), (3), and (4) change with respect to $\alpha, \beta, \delta$ in Appendix A.2. We perform an ablation study on $\alpha, \beta, \delta$ in Appendix A.3 to show their necessity in effectively training with LiLAW. We also show that training with and without LiLAW share the same time complexity and space complexity in Appendix A.4 and Appendix A.5, respectively.

### 4.1 DATASETS

We use the CIFAR-100 dataset (Krizhevsky et al., 2009b) (along with CIFAR-10 (Krizhevsky et al., 2009a), FashionMNIST (Xiao et al., 2017), and MNIST Deng (2012)), but we do not use the full training sets. Instead, we reserve a portion of the training set to serve as the validation set to evaluate and adjust our model's performance, as required by our method. We call our modified datasets, CIFAR-100-M, CIFAR-10-M, FashionMNIST-M, and MNIST-M, respectively.

To demonstrate generalizability and applicability to the medical domain, we also evaluate LiLAW on ten 2D datasets with different medical imaging modalities focusing on multi-class classification tasks from the MedMNISTv2 (Yang et al., 2021; 2023; Doerrich et al., 2024) collection. We also demonstrate strong results on a non-imaging dataset, ECG5000 (PhysioBank, 2000), which contains time-series data for heartbeat classification with 5 classes that are highly imbalanced.

## 4.2 MODELS & IMPLEMENTATION DETAILS

We use models with a varying number of parameters to demonstrate the utility of LiLAW:

– ViT-Base-16-224 (Dosovitskiy et al., 2020), with ImageNet-21K pretraining, on 224×224 inputs with CIFAR-100-M, CIFAR-10-M, FashionMNIST-M, and MNIST-M. We trained for 10 epochs with batch size 16, learning rate 0.0002, and weight decay 0, using a linear learning rate scheduler.
– ResNet-18 (He et al., 2015), with ImageNet-21K pretraining, on 224×224 inputs with the MedM-NISTv2 datasets. We trained for 100 epochs with batch size 128, learning rate 0.0001, and weight decay 0, using a multi-step learning rate scheduler with a 0.1 decay at 50 epochs and 75 epochs, as mentioned in Yang et al. (2023).
– A simple Stacked LSTM model, trained from scratch, on the ECG5000 dataset. We trained for 500 epochs with batch size 512, learning rate 0.001, and weight decay 0.

All models use the Adam optimizer (Kingma, 2014) with early stopping. We use a warmup period of 1 epoch to ensure that the model briefly learns from the data before using LiLAW. The parameters are initialized to $\alpha_{init}, \beta_{init}, \delta_{init} = 10, 2, 6$, with learning rates $\alpha_{lr}, \beta_{lr}, \delta_{lr} = 0.005$, and weight decays $\alpha_{wd}, \beta_{wd}, \delta_{wd} = 0.0001$ for manual gradient descent. Our method is not too sensitive to these choices. We mainly use cross-entropy loss, but also evaluate using focal loss (see Appendix A.10).

Additionally, the CIFAR-100-M, CIFAR-10-M, FashionMNIST-M, and MNIST-M experiments with ViT-Base-16-224, with ImageNet-21K pretraining, on 224x224 inputs were compared with: fine-tuning on the full pretrained model (without LiLAW) and fine-tuning solely on the last 2 layers of the pretrained model (with LiLAW).

## 4.3 EVALUATION

We study the effectiveness of LiLAW with various levels of injected noise ranging from 0%-90% (see Figure 3), with various types of label noise including: uniform, asymmetric, instance, and adjacent; and input noise including: far out-of-distribution, domain shift, covariate shift, zoom shift, and crop shift, as described in Seedat et al. (2024b) (see Table 1), with some MedMNISTv2 datasets (see Table 2 and also see A.14, Figure 6), and with one medical time-series dataset (see Table 3).

In the Appendix, we evaluate LiLAW's performance with ablations on $\alpha, \beta, \delta$ (see A.3, Table 6) to show that including all three parameters achieves the best results, report the results at different noise levels shown in Figure 3 (see A.6, Table 7) to show the effectiveness of LiLAW even in high-noise scenarios, evaluate with and without calibration (using temperature scaling) (see A.7, Table 8) to show the effectiveness of LiLAW regardless of calibration, with and without a clean validation set (see A.8, Table 9) to show that LiLAW does not need a clean validation set and outperforms noise pruning, and with various random seeds (see A.9, Table 10) to show that using LiLAW results in a lower standard deviation in performance. We also show that LiLAW is effective with two different loss functions (see A.10, Table 11), with different validation set sizes (see A.11, Table 12), and with and without the same validation and training distribution (see A.12, Table 13). We report results on four general imaging datasets (see A.13, Table 14), and report the accuracy (see Table 2 and also see A.14, Figure 6) and AUROC (see A.15, Table 15) results for the MedMNISTv2 dataset.

To comprehensively understand LiLAW's impact, we evaluated gains in test accuracy (top-1 and top-5) and AUROC with and without LiLAW across different settings. For the selected MedMNISTv2 datasets, we report the test accuracy (top-1 only, since some have < 5 classes) and AUROC.

## 4.4 PERFORMANCE COMPARISON: EVALUATION ON GENERAL IMAGING DATASETS

In all of the following tables, the results show test performance without LiLAW along with the difference in performance, i.e. the improvement (↑) or deterioration (↓) with LiLAW, respectively.

In Figure 3 (also see A.6, Table 7), we consider the test results at various noise levels from 0% to 90%. At each noise level, LiLAW yields higher test accuracy and AUROC compared to without it.

In Table 1, for each noise type, LiLAW enhances performance even when there is lower overall accuracy without LiLAW. This shows its versatility at effectively handling various noise types. Also, models do not overfit as quickly with LiLAW across all noise levels and all noise types. Although

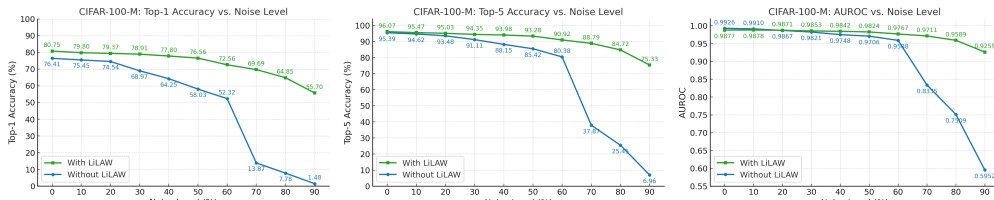

Figure 3: Top-1 accuracy, top-5 accuracy, and AUROC with LiLAW (fine-tuning on the last two layers) and without LiLAW (fine-tuning on the full network) using different levels of uniform noise on CIFAR-100-M. These experiments show the strength of LiLAW even when most of a network's layers are frozen, i.e. LiLAW is effective with partial model fine-tuning.

| Noise Type | Top-1 Acc. (%) | Top-5 Acc. (%) | AUROC |
|---|---|---|---|
| **Label Noise** | | | |
| Uniform | 58.03 ↑ 18.53 | 85.42 ↑ 7.86 | 0.9706 ↑ 0.0118 |
| Asymmetric | 34.37 ↑ 41.43 | 65.75 ↑ 26.97 | 0.9255 ↑ 0.0544 |
| Instance | 60.46 ↑ 6.36 | 88.14 ↑ 1.98 | 0.9787 ↓ 0.0052 |
| Adjacent | 51.80 ↑ 5.60 | 87.74 ↓ 1.51 | 0.9766 ↓ 0.0100 |
| **Input Noise** | | | |
| Far OOD | 34.17 ↑ 5.17 | 48.43 ↑ 1.63 | 0.7739 ↑ 0.0011 |
| Domain Shift | 58.23 ↑ 5.08 | 84.70 ↑ 1.72 | 0.9711 ↑ 0.0097 |
| OOD Covariate | 60.77 ↑ 7.36 | 86.44 ↑ 3.26 | 0.9745 ↓ 0.0014 |
| Zoom Shift | 69.54 ↑ 4.67 | 91.57 ↑ 1.06 | 0.9864 ↑ 0.0091 |
| Crop Shift | 76.35 ↑ 4.67 | 94.82 ↑ 1.07 | 0.9929 ↓ 0.0048 |

Table 1: Results of different noise types at 50% noise level on CIFAR-100-M.

LiLAW was motivated by label noise, the mechanism it learns is source-agnostic. It reweights samples using two observable signals derived from the model's predictions: prediction confidence and model agreement with the observed label. Input noise manifests in these same signals by decreasing confidence and/or decreasing agreement. Because LiLAW's weights are functions of these two quantities and are meta-learned on the validation set after every mini-batch, the method automatically adapts its emphasis to either kind of noise without needing to know which is present.

## 4.5 PERFORMANCE COMPARISON: EVALUATION ON MEDICAL DATASETS

| | Acc. (%) | | | | | |
|---|---|---|---|---|---|---|
| Dataset | 0% Noise | 10% Noise | 20% Noise | 30% Noise | 40% Noise | 50% Noise |
| PathMNIST | 95.49 ↑ 0.08 | 95.38 ↓ 0.08 | 94.70 ↑ 0.20 | 94.19 ↓ 0.59 | 94.01 ↑ 0.18 | 92.87 ↑ 0.69 |
| DermaMNIST | 79.88 ↑ 0.88 | 76.12 ↑ 1.80 | 74.38 ↑ 2.13 | 73.92 ↑ 0.27 | 71.89 ↑ 1.46 | 68.03 ↑ 4.52 |
| OCTMNIST | 75.73 ↑ 0.65 | 72.84 ↑ 4.74 | 71.42 ↑ 1.97 | 67.11 ↑ 3.44 | 68.19 ↑ 4.94 | 60.79 ↑ 7.07 |
| PneumoniaMNIST | 89.04 ↑ 2.61 | 85.40 ↑ 1.10 | 86.65 ↓ 0.41 | 85.40 ↑ 4.18 | 81.50 ↑ 2.90 | 84.24 ↑ 2.14 |
| BreastMNIST | 85.83 ↑ 1.39 | 83.87 ↓ 1.17 | 79.13 ↑ 2.96 | 78.74 ↑ 0.39 | 74.78 ↑ 6.75 | 70.10 ↑ 4.34 |
| BloodMNIST | 98.50 ↑ 0.16 | 97.24 ↑ 0.18 | 97.36 ↑ 0.21 | 96.93 ↓ 0.28 | 95.72 ↑ 0.71 | 94.72 ↑ 0.42 |
| TissueMNIST | 68.74 ↓ 1.33 | 65.02 ↑ 0.85 | 61.33 ↑ 1.17 | 45.08 ↑ 19.16 | 38.50 ↑ 13.22 | 31.82 ↑ 20.41 |
| OrganAMNIST | 94.87 ↑ 0.18 | 94.19 ↑ 0.22 | 94.02 ↑ 0.18 | 90.76 ↑ 2.73 | 92.50 ↑ 0.52 | 89.45 ↑ 1.63 |
| OrganCMNIST | 90.66 ↑ 0.93 | 88.85 ↑ 1.01 | 85.37 ↑ 0.30 | 84.45 ↑ 2.73 | 84.19 ↑ 0.87 | 82.95 ↑ 1.24 |
| OrganSMNIST | 80.54 ↑ 0.49 | 78.28 ↑ 0.66 | 75.34 ↑ 2.16 | 72.54 ↑ 3.35 | 72.95 ↑ 1.42 | 70.82 ↑ 2.65 |

Table 2: Accuracy on ten 2D datasets from MedMNISTv2 with different levels of uniform noise. The MedMNIST experiments were done with fine-tuning on the full pretrained model without LiLAW and with LiLAW, i.e. LiLAW is effective with full model fine-tuning.

In Table 2 (also see A.14, Figure 6), we see the effect of LiLAW across ten 2D datasets from MedMNISTv2 with varying levels of injected uniform label noise (see A.15, Table 15, Figure 7 for AUROC metrics). Applying LiLAW across various uniform noise levels leads to an increase in accuracy and/or AUROC in nearly all cases compared to when trained without LiLAW. When there is deterioration, it is minimal and usually takes place at lower noise levels. Given the noisiness of

medical data, LiLAW's ability to enhance performance under such conditions is highly valuable, especially given that we are also performing transfer learning with ImageNet-21K pretraining.

| Dataset | Acc. (%) | AUROC |
|---------|----------|-------|
| ECG5000 | 93.60 ↑ 3.20 | 0.9982 ↑ 0.0005 |

Table 3: Results on the ECG5000 dataset.

Table 3 highlights the effectiveness of LiLAW on ECG5000. Compared to standard training, LiLAW yields a substantial improvement in accuracy and AUROC, demonstrating its ability to enhance predictive performance and discrimination in inherently noisy domains. Namely, LiLAW can dynamically reweight samples based on difficulty to effectively mitigate the impact of inherent noise in physiological signals. This suggests that LiLAW provides a generalizable framework to improve robustness across modalities where label quality and data heterogeneity are common challenges.

### 4.6 PERFORMANCE COMPARISON: IDENTIFYING MISLABELS

LiLAW's high AUCROC and AUCPRC values suggest that it is highly effective at identifying mislabeled samples at early-stage and late-stage training and consistently outperforms several methods for difficulty estimation, including Data-IQ (Seedat et al., 2022), DataMaps (Swayamdipta et al., 2020), CNLCU-S (Xia et al., 2021), AUM (Pleiss et al., 2020), EL2N (Paul et al., 2023), Grand (Paul et al., 2023), and Forgetting (Toneva et al., 2019) (see A.16, Table 16).

### 4.7 PERFORMANCE COMPARISON: NOISY-LABEL LEARNING

Table 4: Comparison of methods on noisy-label learning on the Clothing-1M dataset.

| Method | Accuracy (%) |
|--------|--------------|
| Cross-entropy | 69.21 |
| Backward (Patrini et al., 2016) | 69.13 |
| Forward (Patrini et al., 2016) | 69.84 |
| Joint-Optim (Tanaka et al., 2018) | 72.16 |
| MetaCleaner (Zhang et al., 2019) | 72.50 |
| GCE (Zhang & Sabuncu, 2018) | 69.75 |
| SL (Wang et al., 2019) | 71.02 |
| LiLAW (ours) | 71.24 |

Using the Clothing-1M dataset (Xiao et al., 2015), we evaluate ResNet-50 with ImageNet-21K pretraining under a real-world noisy dataset. The same training method used in the methods mentioned in Table 4 (results obtained from the respective papers) was used with LiLAW. Several noisy-label learning methods, including meta-learning methods, improve performance over standard cross-entropy, confirming their effectiveness in mitigating label corruption. Some of the stronger baselines require substantially heavier training pipelines, but in contrast, our method achieves 71.24% accuracy. This makes our method competitive with, and in several cases, exceed baseline performance while maintaining a lightweight design without additional networks or additional prediction heads. This suggests that our method is a practical and scalable choice for real-world noisy-label settings.

## 5 CONCLUSION

In this work, we introduced LiLAW, a simple and lightweight adaptive weighting method that consistently improves training under noisy and heterogeneous conditions. By learning only three parameters that evolve with sample difficulty, LiLAW can be easily integrated into existing pipelines without added complexity. We highlight LiLAW's practical utility using medical imaging and time-series data. In future work, this framework can be extended to regression, multi-label classification, and dense prediction tasks. LiLAW also opens up opportunities for active learning (select difficult samples), continual learning (stabilize when there is drift), semi-supervised learning (downweigh low-confidence pseudo-labels), and bias mitigation (upweigh underrepresented cohorts), while handling class imbalance more adaptively than static methods.

ETHICS STATEMENT

All datasets used in this work are publicly available and widely used for research purposes. This work does not present any known ethical concerns.

REPRODUCIBILITY STATEMENT

The details provided about Datasets (see Section 4.1) and Models & Implementation (see Section 4.2) along with the code provided in the Supplementary Material make this work reproducible.

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

# A APPENDIX

## A.1 MOTIVATING EXAMPLE

| Case (Label is [1,0]) | Prediction | $s_y$ | $s_{max}$ | CE | $\alpha=10,\ \beta=2,\ \delta=6$ | | | | | $\alpha=9,\ \beta=3,\ \delta=7$ | | | | |
|---|---|---|---|---|---|---|---|---|---|---|---|---|---|---|
| | | | | | $W_\alpha$ | $W_\delta$ | $W_\beta$ | $W$ | $\mathcal{L}_W$ | $W_\alpha$ | $W_\delta$ | $W_\beta$ | $W$ | $\mathcal{L}_W$ |
| Correct & Confident | [0.95,0.05] | 0.95 | 0.95 | **0.051** | 0.999 | 0.199 | 0.500 | 1.698 | **0.087** | 0.999 | 0.000 | 0.130 | 1.129 | **0.058** |
| Correct & Unconfident | [0.60,0.40] | 0.60 | 0.60 | **0.511** | 0.998 | 0.865 | 0.500 | 2.363 | **1.208** | 0.992 | 0.002 | 0.231 | 1.225 | **0.626** |
| Incorrect & Unconfident | [0.40,0.60] | 0.40 | 0.60 | **0.916** | 0.982 | 0.966 | 0.550 | 2.498 | **2.289** | 0.953 | 0.089 | 0.354 | 1.396 | **1.279** |
| Incorrect & Confident | [0.05,0.95] | 0.05 | 0.95 | **2.996** | 0.222 | 0.658 | 0.812 | 1.692 | **5.073** | 0.378 | 0.835 | 0.690 | 1.903 | **5.700** |

Table 5: Comparison of cross-entropy (CE) and LiLAW-weighted loss under two $\alpha, \beta, \delta$ settings.

As shown in Table 5, LiLAW adapts weighting beyond what CE alone provides. When predictions are correct and confident, LiLAW assigns relatively small weights, keeping the loss close to CE and preventing overfitting. When predictions are correct but unconfident, LiLAW amplifies the loss and gives importance to samples that CE might undervalue. When predictions are incorrect but unconfident, LiLAW boosts the loss to ensure that the model learns from these ambiguous samples. When predictions are incorrect and confident, LiLAW increases the penalty, discouraging the network from becoming overconfident in wrong answers. The main benefit of LiLAW comes from the dynamic nature of $\alpha, \beta, \delta$. If $\alpha = 10 \rightarrow 9$, $\beta = 2 \rightarrow 3$, and $\delta = 6 \rightarrow 7$, then whereas CE still has the same penalties for the cases if they are seen again, LiLAW further decreases the penalty on the correct cases and unconfident cases, while increasing the penalty on the incorrect and confident cases, which causes the incorrect and confident cases to self-correct more effectively.

## A.2 DERIVATIVES OF WEIGHT FUNCTIONS

We consider the derivatives of our weight functions based on $\alpha, \beta, \delta$ with respect to the LiLAW weighted loss function to study how they grow with those three parameters. Note that our weight functions are defined as in the Method section (see Section 3):

$$\mathcal{W}_\alpha(s_i, \widetilde{y_i}) = \sigma(\alpha \cdot s_i[\widetilde{y_i}] - \max(s_i))$$

$$\mathcal{W}_\beta(s_i, \widetilde{y_i}) = \sigma(-(\beta \cdot s_i[\widetilde{y_i}] - \max(s_i)))$$

$$\mathcal{W}_\delta(s_i, \widetilde{y_i}) = \exp\left(-\frac{(\delta \cdot s_i[\widetilde{y_i}] - \max(s_i))^2}{2}\right)$$

We calculated the weight for each sample as follows:

$$\mathcal{W}(s_i, \widetilde{y_i}) = \mathcal{W}_\alpha(s_i, \widetilde{y_i}) + \mathcal{W}_\beta(s_i, \widetilde{y_i}) + \mathcal{W}_\delta(s_i, \widetilde{y_i})$$

and defined our LiLAW weighted loss function as follows:

$$\mathcal{L}_W = \mathcal{W}(s_i, \widetilde{y_i}) \cdot \mathcal{L}$$

Based on the above definitions, we have the following derivatives:

$$\nabla_\alpha \mathcal{L}_W = \frac{\partial \mathcal{L}_W}{\partial \alpha} = \frac{\partial}{\partial \alpha}(\mathcal{W}_\alpha(s_i, \widetilde{y_i}) \cdot \mathcal{L}) = \mathcal{L} \cdot \frac{\partial}{\partial \alpha}\mathcal{W}_\alpha(s_i, \widetilde{y_i})$$

$$= \mathcal{L} \cdot \frac{\partial}{\partial \alpha}(\sigma(\alpha \cdot s_i[\widetilde{y_i}] - \max(s_i)))$$

$$= \frac{\mathcal{L} \cdot (\sigma(\alpha \cdot s_i[\widetilde{y_i}] - \max(s_i)))^2 \cdot s_i[\widetilde{y_i}]}{\exp\left(\alpha \cdot s_i[\widetilde{y_i}] - \max(s_i)\right)}$$

$$= \frac{\mathcal{L} \cdot \mathcal{W}_\alpha(s_i, \widetilde{y_i})^2 \cdot s_i[\widetilde{y_i}]}{\exp\left(\alpha \cdot s_i[\widetilde{y_i}] - \max(s_i)\right)}$$

Note: $\nabla_\alpha \mathcal{L}_W \geq 0$ as $\mathcal{L} \geq 0$, $\mathcal{W}_\alpha(s_i, \widetilde{y_i})^2 > 0$, $s_i[\widetilde{y_i}] \geq 0$, and $\exp\left(\alpha \cdot s_i[\widetilde{y_i}] - \max(s_i)\right) > 0$.

$$\nabla_\beta \mathcal{L}_W = \frac{\partial \mathcal{L}_W}{\partial \beta} = \frac{\partial}{\partial \beta}(\mathcal{W}_\beta(s_i, \widetilde{y_i}) \cdot \mathcal{L}) = \mathcal{L} \cdot \frac{\partial}{\partial \beta} \mathcal{W}_\beta(s_i, \widetilde{y_i})$$

$$= \mathcal{L} \cdot \frac{\partial}{\partial \beta}(\sigma(-(\beta \cdot s_i[\widetilde{y_i}] - \max(s_i))))$$

$$= -\frac{\mathcal{L} \cdot (\sigma(-(\beta \cdot s_i[\widetilde{y_i}] - \max(s_i))))^2 \cdot s_i[\widetilde{y_i}]}{\exp\left(-(\beta \cdot s_i[\widetilde{y_i}] - \max(s_i))\right)}$$

$$= -\frac{\mathcal{L} \cdot \mathcal{W}_\beta(s_i, \widetilde{y_i})^2 \cdot s_i[\widetilde{y_i}]}{\exp\left(-(\beta \cdot s_i[\widetilde{y_i}] - \max(s_i))\right)}$$

Note: $\nabla_\beta \mathcal{L}_W \leq 0$ as $\mathcal{L} \geq 0$, $\mathcal{W}_\beta(s_i, \widetilde{y_i})^2 > 0$, $s_i[\widetilde{y_i}] \geq 0$, and $\exp\left(-(\beta \cdot s_i[\widetilde{y_i}] - \max(s_i))\right) > 0$.

$$\nabla_\delta \mathcal{L}_W = \frac{\partial \mathcal{L}_W}{\partial \delta} = \frac{\partial}{\partial \delta}(\mathcal{W}_\delta(s_i, \widetilde{y_i}) \cdot \mathcal{L}) = \mathcal{L} \cdot \frac{\partial}{\partial \delta} \mathcal{W}_\delta(s_i, \widetilde{y_i})$$

$$= \mathcal{L} \cdot \frac{\partial}{\partial \delta}\left(\exp\left(-\frac{(\delta \cdot s_i[\widetilde{y_i}] - \max(s_i))^2}{2}\right)\right)$$

$$= -\frac{\mathcal{L} \cdot (\delta \cdot s_i[\widetilde{y_i}] - \max(s_i)) \cdot s_i[\widetilde{y_i}]}{\exp\left(\frac{(\delta \cdot s_i[\widetilde{y_i}] - \max(s_i))^2}{2}\right)}$$

$$= -\mathcal{L} \cdot \mathcal{W}_\delta(s_i, \widetilde{y_i}) \cdot (\delta \cdot s_i[\widetilde{y_i}] - \max(s_i)) \cdot s_i[\widetilde{y_i}]$$

Note: $\mathcal{L} \geq 0$, $\mathcal{W}_\delta(s_i, \widetilde{y_i})^2 > 0$, and $s_i[\widetilde{y_i}] \geq 0$, we see that $\nabla_\delta \mathcal{L}_W \leq 0$ when $\delta \cdot s_i[\widetilde{y_i}] \geq \max(s_i)$ and $\nabla_\delta \mathcal{L}_W > 0$ when $\delta \cdot s_i[\widetilde{y_i}] < \max(s_i)$.

In summary, the gradients for the parameters are updated using autograd, but we show why $\alpha$ always decreases and $\beta$ always increases at different rates. On the other hand, $\delta$ could increase or decrease depending on the conditions above. This is the reason for our choice of high $\alpha$, low $\beta$, and $\delta$ in between (by default, $\alpha = 10, \beta = 2, \delta = 6$). Figure 4 shows that $\alpha$ decreases, $\beta$ increases, and $\delta$ increases as discussed above. The reason $\alpha$ decreases faster, $\delta$ increases faster, and $\beta$ increases faster with 50% noise than 0% noise is because easy samples are less reliable sooner, moderate cases are more informative faster, and hard samples are also more informative faster, respectively.

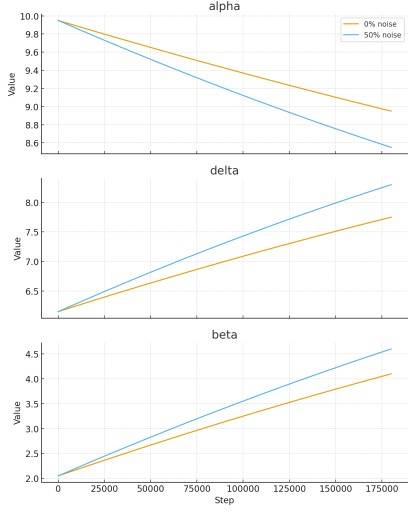

Figure 4: Plots showing how $\alpha, \beta, \delta$ change over the course of training with 0% uniform noise and 50% uniform noise on CIFAR-100-M.

## A.3 ABLATION STUDY ON $\alpha, \beta, \delta$

| Parameters Used | Noise Level (%) | Top-1 Acc. (%) | Top-5 Acc. (%) | AUROC |
|---|---|---|---|---|
| $\alpha, \beta, \delta$ | 0 | 76.41 ↑ 4.34 | 95.39 ↑ 0.68 | 0.9926 ↓ 0.0049 |
| | 50 | 58.03 ↑ 18.53 | 85.42 ↑ 7.86 | 0.9706 ↑ 0.0118 |
| $\alpha, \beta$ | 0 | 76.41 ↑ 4.37 | 95.39 ↑ 0.74 | 0.9926 ↓ 0.0040 |
| | 50 | 58.03 ↑ 17.52 | 85.42 ↑ 7.26 | 0.9706 ↑ 0.0106 |
| $\alpha, \delta$ | 0 | 76.41 ↑ 4.45 | 95.39 ↑ 0.67 | 0.9926 ↓ 0.0046 |
| | 50 | 58.03 ↑ 16.48 | 85.42 ↑ 6.88 | 0.9706 ↑ 0.0093 |
| $\beta, \delta$ | 0 | 76.41 ↑ 2.85 | 95.39 ↑ 0.39 | 0.9926 ↓ 0.0061 |
| | 50 | 58.03 ↑ 17.03 | 85.42 ↑ 7.54 | 0.9706 ↑ 0.0110 |
| $\alpha$ | 0 | 76.41 ↑ 3.34 | 95.39 ↑ 0.05 | 0.9926 ↓ 0.0039 |
| | 50 | 58.03 ↑ 16.49 | 85.42 ↑ 6.89 | 0.9706 ↑ 0.0098 |
| $\beta$ | 0 | 76.41 ↑ 3.21 | 95.39 ↑ 0.47 | 0.9926 ↓ 0.0049 |
| | 50 | 58.03 ↑ 16.48 | 85.42 ↑ 6.88 | 0.9706 ↑ 0.0098 |
| $\delta$ | 0 | 76.41 ↑ 2.54 | 95.39 ↑ 0.42 | 0.9926 ↓ 0.0064 |
| | 50 | 58.03 ↑ 16.49 | 85.42 ↑ 6.89 | 0.9706 ↑ 0.0098 |

Table 6: Results of ablating the parameters with different levels of uniform noise on CIFAR-100-M.

In Table 6, we present an ablation study examining the impact of different combinations of the LiLAW parameters $(\alpha, \beta, \delta)$ on model performance under varying noise levels. Using all three parameters yields the highest performance gains, which are especially significant at higher noise levels. This demonstrates the full potential of LiLAW in improving model robustness and accuracy. Excluding one or two of the weights still results in improvements in the performance, however including all three yields the best results.

## A.4 TIME COMPLEXITY ANALYSIS

Without LiLAW, the runtime for each batch in each epoch in Algorithm 1 is $O(|\theta| \cdot B)$ for the forward pass, loss calculation, backward pass, and update step, where $\theta$ are the model parameters and $B$ is the batch size. Going through all batches, we traverse the full dataset, so the runtime for each epoch is $O(|\theta| \cdot |\mathcal{D}_t + \mathcal{D}_v|)$. The total for $n$ epochs is $O(n \cdot |\theta| \cdot |\mathcal{D}_t + \mathcal{D}_v|)$.

With LiLAW, the runtime for the forward pass, loss calculation, backward pass, and update step is $O((|\theta| + 3) \cdot B) = O(|\theta| \cdot B)$ for each batch in each epoch in Algorithm 1, where $\theta$ are the model parameters, $B$ is the batch size (assuming the same batch size for training and validation), and 3 refers to the three LiLAW parameters. Going through all training batches and a single validation batch after each training batch, we have $O(|\theta| \cdot (|\mathcal{D}_t + \mathcal{D}_v| \cdot B)) = O(|\theta| \cdot |\mathcal{D}_t + \mathcal{D}_v|)$ as the runtime for each epoch. The total for $n$ epochs is $O(n \cdot |\theta| \cdot |\mathcal{D}_t + \mathcal{D}_v|)$, same as without LiLAW.

## A.5 SPACE COMPLEXITY ANALYSIS

Without LiLAW, the space complexity for Algorithm 1 is $O(|\theta|)$ for the model parameters $\theta$, $O(|\theta|)$ for the model parameter gradients, and $O(|\theta| \cdot B)$ for the activations during the forward pass, where $B$ is the batch size. The total is therefore $O(|\theta| \cdot B)$.

With LiLAW, the space complexity for Algorithm 1 is $O(|\theta| + 3) = O(|\theta|)$ for the model parameters, $\theta$, and the three LiLAW parameters, $\alpha, \beta, \delta$, $O(|\theta| + 3) = O(|\theta|)$ for the model parameter gradients and the LiLAW parameter gradients, and $O((|\theta| + 3) \cdot B) = O(|\theta| \cdot B)$ for the activations during the forward pass, where $B$ is the batch size (assuming the same batch size for training and validation). The total is $O(|\theta| \cdot B)$, same as without LiLAW.

## A.6 PERFORMANCE WITH VARIOUS NOISE LEVELS

| Noise Level (%) | Top-1 Acc. (%) | Top-5 Acc. (%) | AUROC |
|---|---|---|---|
| 0 | 76.41 ↑ 4.34 | 95.39 ↑ 0.68 | 0.9926 ↓ 0.0049 |
| 10 | 75.45 ↑ 4.35 | 94.62 ↑ 0.85 | 0.9910 ↓ 0.0032 |
| 20 | 74.54 ↑ 4.83 | 93.48 ↑ 1.55 | 0.9867 ↑ 0.0004 |
| 30 | 68.97 ↑ 9.94 | 91.11 ↑ 3.24 | 0.9821 ↑ 0.0032 |
| 40 | 64.25 ↑ 13.55 | 88.15 ↑ 5.83 | 0.9748 ↑ 0.0094 |
| 50 | 58.03 ↑ 18.53 | 85.42 ↑ 7.86 | 0.9706 ↑ 0.0118 |
| 60 | 52.32 ↑ 20.24 | 80.38 ↑ 10.54 | 0.9588 ↑ 0.0179 |
| 70 | 13.87 ↑ 55.82 | 37.87 ↑ 50.92 | 0.8335 ↑ 0.1376 |
| 80 | 7.78 ↑ 57.07 | 25.45 ↑ 59.27 | 0.7509 ↑ 0.2080 |
| 90 | 1.48 ↑ 54.22 | 6.96 ↑ 68.37 | 0.5952 ↑ 0.3307 |

Table 7: Results with different levels of uniform noise on CIFAR-100-M.

In Table 7, we report test performance across noise levels from 0% to 90% in increments of 10%. At every setting, LiLAW consistently improves both accuracy and AUROC over the baseline. Notably, with LiLAW, top-1 accuracy under up to 50% noise matches that of noiseless training.

## A.7 PERFORMANCE WITH AND WITHOUT CALIBRATION

| Calibration | Noise Level (%) | Top-1 Acc. (%) | Top-5 Acc. (%) | AUROC |
|---|---|---|---|---|
| Without calibration | 0 | 76.41 ↑ 4.34 | 95.39 ↑ 0.68 | 0.9926 ↓ 0.0049 |
| | 50 | 58.03 ↑ 18.53 | 85.42 ↑ 7.86 | 0.9706 ↑ 0.0118 |
| With calibration | 0 | 76.24 ↑ 4.49 | 95.34 ↑ 0.71 | 0.9923 ↓ 0.0046 |
| | 50 | 58.43 ↑ 18.09 | 84.32 ↑ 8.96 | 0.9715 ↑ 0.0109 |

Table 8: Results with and without calibration with different levels of uniform noise on CIFAR-100-M.

According to Table 8, with and without noise, applying LiLAW leads to significant performance gains, regardless of calibration. When calibration is combined with LiLAW, it works synergistically to enhance robustness to noise. The performance boost at 50% noise indicates that LiLAW effectively mitigates the effects of label noise. Using LiLAW with 50% noise surpasses the top-1 accuracy of not using LiLAW with 0% noise. We note that there is a reasonable boost in test accuracy when using LiLAW even when there is 0% noise since we are pushing unconfident predictions to be more confident. There is a slight decrease in AUROC in nearly all cases where there is 0% noise since the model may need to be slightly less confident on the thresholds to improve accuracy.

## A.8 PERFORMANCE WITH AND WITHOUT A CLEAN VALIDATION SET

| Validation set cleanliness | Noise Level (%) | Top-1 Acc. (%) | Top-5 Acc. (%) | AUROC |
|---|---|---|---|---|
| Without a clean validation set | 0 | 76.41 ↑ 4.34 | 95.39 ↑ 0.68 | 0.9926 ↓ 0.0049 |
| | 50 | 58.03 ↑ 18.53 | 85.42 ↑ 7.86 | 0.9706 ↑ 0.0118 |
| With a clean validation set | 0 | 77.40 ↑ 3.33 | 95.56 ↑ 0.48 | 0.9934 ↓ 0.0057 |
| | 50 | 55.42 ↑ 21.07 | 84.72 ↑ 8.59 | 0.9734 ↑ 0.0090 |
| Trained on 50% of data that is clean | 0 | 74.34 | 94.94 | 0.9919 |

Table 9: Results with and without a clean validation set on CIFAR-100-M with 50% uniform noise. We also compare these results to when the model is trained only on the 50% of the data that is clean.

In Table 9, we show test performance with and without a clean validation set. We see that LiLAW significantly enhances performance even without a clean validation set, demonstrating its robustness in practical scenarios where obtaining a clean validation set may be challenging. The improvements with a clean validation set are comparable to those without one, indicating that LiLAW does not heavily rely on validation set cleanliness and can easily adapt to noisy validation data. In addition, training on only 50% of the data that is clean (without using the 50% noisy data) achieves worse performance than using LiLAW without a clean validation set, further supporting our claim that LiLAW is robust to noise and more effective than pruning noisy data.

## A.9 PERFORMANCE UNDER DIFFERENT RANDOM SEEDS

Table 10 and Figure 5 shows that the improvements observed with LiLAW are consistent across five different random initializations and choices of meta-validation sets, as reflected by the low standard deviations over five independent runs. Under 0% noise and 50% noise, LiLAW not only achieves much higher accuracy but also reduces variability across the runs. This demonstrates that LiLAW yields reliable performance improvements across various training conditions, highlighting its robustness to noise and stochasticity.

| Noise Level (%) | LiLAW | Top-1 Acc. (%) | Top-5 Acc. (%) | AUROC |
|---|---|---|---|---|
| 0 | ✗ | $74.93 \pm 1.07$ | $94.56 \pm 0.51$ | $0.9918 \pm 0.0009$ |
| | ✓ | $80.87 \pm 0.27$ | $95.91 \pm 0.08$ | $0.9881 \pm 0.0003$ |
| 50 | ✗ | $46.32 \pm 7.82$ | $76.12 \pm 5.98$ | $0.9537 \pm 0.0142$ |
| | ✓ | $75.68 \pm 1.03$ | $92.91 \pm 0.53$ | $0.9820 \pm 0.0012$ |

Table 10: Performance (mean $\pm$ std over 5 runs with different random seeds) on CIFAR-100-M across different noise levels with (✓) and without (✗) LiLAW.

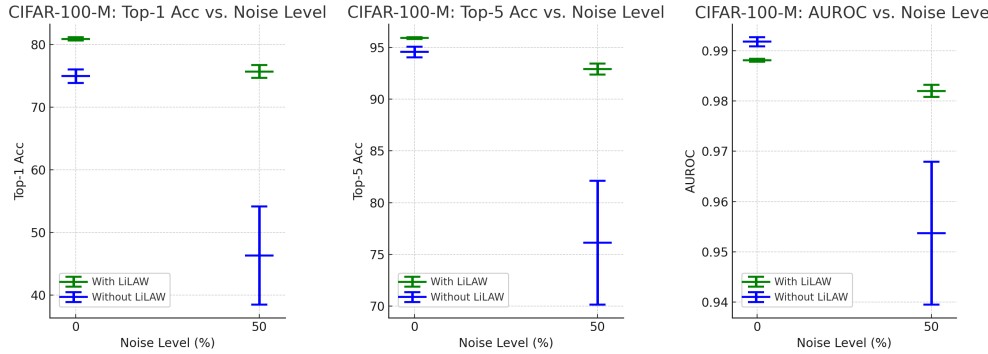

Figure 5: Plots with means and standard deviations of top-1 accuracy, top-5 accuracy, and AUROC across 5 runs with different random seeds on CIFAR-100-M under different noise levels.

## A.10 PERFORMANCE UNDER DIFFERENT LOSS FUNCTIONS

| Loss Function | Noise Level (%) | Top-1 Acc. (%) | Top-5 Acc. (%) | AUROC |
|---|---|---|---|---|
| Cross-Entropy | 0 | 76.41 ↑ 4.34 | 95.39 ↑ 0.68 | 0.9926 ↓ 0.0049 |
| | 50 | 58.03 ↑ 18.53 | 85.42 ↑ 7.86 | 0.9706 ↑ 0.0118 |
| Focal Loss (Lin et al., 2017) | 0 | 78.51 ↑ 1.46 | 96.70 ↓ 0.97 | 0.9940 ↓ 0.0063 |
| | 50 | 56.66 ↑ 19.76 | 85.36 ↑ 7.96 | 0.9746 ↑ 0.0079 |

Table 11: Results of two loss functions with varying levels of uniform noise on CIFAR-100-M.

In Table 11, we see that LiLAW provides performance gains with both cross-entropy and focal loss functions, indicating its versatility even when we use different loss landscapes that are already designed to handle issues such as class imbalance. Although noisy labels can negatively affect training with either of these two losses, LiLAW's adaptive weighting helps mitigate the impact of mislabeled or noisy data by dynamically adjusting the loss weight of each sample. Note that the boosts with LiLAW with cross-entropy loss and with focal loss reach similar accuracies.

## A.11 EFFECT OF VALIDATION SET SIZE

In Table 12, we analyze the effect of varying the validation set size (as a percentage of the training set size) on the test performance. We see that we do not need too much validation data to obtain high performance using LiLAW. Note that we only use one random batch from the validation set. We see that a 15% validation set size strikes a good balance between having enough validation data for LiLAW while leaving sufficient data for model training. Simply increasing the validation set size

| Validation Set Size (%) | Top-1 Acc. (%) | Top-5 Acc. (%) | AUROC |
|:---:|:---:|:---:|:---:|
| 5 | 47.43 ↑ 27.61 | 76.95 ↑ 15.63 | 0.9513 ↑ 0.0288 |
| 10 | 42.58 ↑ 31.87 | 72.63 ↑ 19.76 | 0.9445 ↑ 0.0361 |
| 15 | 58.03 ↑ 18.53 | 85.42 ↑ 7.86 | 0.9706 ↑ 0.0118 |
| 20 | 38.21 ↑ 37.45 | 70.51 ↑ 22.77 | 0.9417 ↑ 0.0407 |
| 25 | 59.32 ↑ 17.38 | 85.20 ↑ 7.49 | 0.9679 ↑ 0.0125 |
| 30 | 61.66 ↑ 14.65 | 85.75 ↑ 7.07 | 0.9717 ↑ 0.0091 |

Table 12: Results with different validation set sizes (as a % of the training set size) on CIFAR-100-M with 50% uniform noise.

does not guarantee better performance. As a result, we conclude that there is minor variability in the boost from LiLAW depending on the validation set size, but LiLAW consistently improves accuracy across all validation set sizes, demonstrating its robustness in noisy settings.

## A.12 EFFECT OF VALIDATION SET DISTRIBUTION

| Validation Set Distribution | Noise Level (%) | Top-1 Acc. (%) | Top-5 Acc. (%) | AUROC |
|:---|:---:|:---:|:---:|:---:|
| Same distribution | 0 | 76.41 ↑ 4.34 | 95.39 ↑ 0.68 | 0.9926 ↓ 0.0049 |
| | 50 | 58.03 ↑ 18.53 | 85.42 ↑ 7.86 | 0.9706 ↑ 0.0118 |
| Different distribution | 0 | 76.41 ↑ 3.64 | 95.39 ↑ 0.12 | 0.9926 ↓ 0.0045 |
| | 50 | 58.03 ↑ 18.50 | 85.42 ↑ 7.87 | 0.9706 ↑ 0.0118 |

Table 13: Results with a validation set from the same distribution as the training set and with a validation set from a different distribution than the training set (through random flipping, rotations, and color jitter augmentations) on CIFAR-100-M with 50% uniform noise.

In Table 13, we explore the effect of the validation set distribution on test performance. Namely, we compare a validation set drawn from the same distribution as the training set versus one drawn from a different distribution (created through augmentations like random flipping, rotations, and color jitter). Results indicate that LiLAW is slightly more effective when the validation set matches the training set distribution, though the difference is relatively small. We conclude that the distribution of the validation set has a minor impact on the effectiveness of LiLAW and that LiLAW remains robust even when the validation set distribution differs, which may be useful in cases where all the training data has to be used for training. This suggests that while matching the validation set distribution to the training set is ideal, LiLAW can still provide significant improvements in noisy settings even when this condition is not perfectly met.

## A.13 RESULTS ON ADDITIONAL GENERAL IMAGING DATASETS

| Dataset | Noise Level (%) | Top-1 Acc. (%) | Top-5 Acc. (%) | AUROC |
|:---|:---:|:---:|:---:|:---:|
| MNIST-M | 0 | 97.16 ↑ 0.78 | 99.97 ↑ 0.01 | 0.9984 ↓ 0.0038 |
| | 50 | 77.52 ↑ 18.37 | 98.32 ↑ 1.57 | 0.9822 ↑ 0.0139 |
| FashionMNIST-M | 0 | 89.67 ↑ 1.11 | 99.85 ↑ 0.11 | 0.9855 ↑ 0.0030 |
| | 50 | 79.41 ↑ 8.35 | 98.29 ↑ 1.28 | 0.9751 ↑ 0.0105 |
| CIFAR-10-M | 0 | 92.15 ↑ 2.51 | 99.80 ↓ 0.02 | 0.9942 ↓ 0.0004 |
| | 50 | 85.65 ↑ 7.05 | 98.15 ↑ 1.16 | 0.9787 ↑ 0.0118 |
| CIFAR-100-M | 0 | 76.41 ↑ 4.34 | 95.39 ↑ 0.68 | 0.9926 ↓ 0.0049 |
| | 50 | 58.03 ↑ 18.53 | 85.42 ↑ 7.86 | 0.9706 ↑ 0.0118 |

Table 14: Results on various datasets with different levels of uniform noise.

In Table 14, we note that LiLAW improves accuracy on all datasets (with only a minor drop in top-5 accuracy when there is 0% noise in the CIFAR-10-M dataset). In nearly all cases, the improvement with LiLAW when there is 50% noise is close to the performance without LiLAW when there is 0% noise. This suggests that LiLAW is beneficial for both simple and complex classification tasks.

## A.14 ACCURACY PLOTS FOR MEDMNISTV2

Figure 6 shows LiLAW's impact on ten MedMNISTv2 datasets under varying levels of uniform noise. Across nearly all settings, we see an improvement with LiLAW in the accuracy and/or the AUROC compared to the baseline, with only minor drops occurring occasionally at lower noise levels.

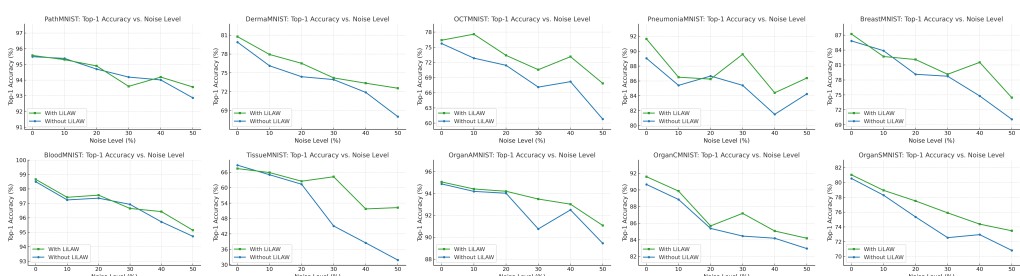

Figure 6: Accuracy with and without LiLAW on ten 2D datasets from MedMNISTv2.

## A.15 AUROC FOR MEDMNISTV2

In Table 15 and Figure 7, we present the AUCROC for ten 2D medical imaging datasets from MedMNISTv2 at varying levels of label noise. Across most datasets and noise levels, LiLAW enhances the AUCROC scores, ranging from minor to substantial gains. A few instances show slight decreases in AUCROC when LiLAW is applied. However, in general, LiLAW provides modest improvements or minor deteriorations and in all noise levels.

| Dataset | AUROC | | | | | |
| --- | --- | --- | --- | --- | --- | --- |
| | 0% Noise | 10% Noise | 20% Noise | 30% Noise | 40% Noise | 50% Noise |
| PathMNIST | 0.9968 ↑ 0.0001 | 0.9952 ↑ 0.0009 | 0.9920 ↑ 0.0011 | 0.9811 ↑ 0.0014 | 0.9873 ↑ 0.0012 | 0.9887 ↑ 0.0015 |
| DermaMNIST | 0.9206 ↑ 0.0095 | 0.8647 ↑ 0.0026 | 0.8606 ↓ 0.0179 | 0.8322 ↑ 0.0070 | 0.7930 ↓ 0.0153 | 0.7712 ↑ 0.0039 |
| OCTMNIST | 0.9925 ↑ 0.0004 | 0.9757 ↑ 0.0082 | 0.9840 ↓ 0.0013 | 0.9727 ↑ 0.0097 | 0.9649 ↑ 0.0128 | 0.9289 ↑ 0.0451 |
| PneumoniaMNIST | 0.9803 ↑ 0.0049 | 0.9700 ↓ 0.0064 | 0.9536 ↑ 0.0209 | 0.9123 ↑ 0.0237 | 0.8758 ↑ 0.0238 | 0.9424 ↑ 0.0058 |
| BreastMNIST | 0.8638 ↑ 0.0089 | 0.8620 ↑ 0.0043 | 0.8549 ↑ 0.0074 | 0.8065 ↑ 0.0099 | 0.7424 ↑ 0.0150 | 0.7562 ↓ 0.0010 |
| BloodMNIST | 0.9990 ↑ 0.0001 | 0.9980 ↑ 0.0002 | 0.9981 ↑ 0.0003 | 0.9974 ↓ 0.0002 | 0.9953 ↑ 0.0010 | 0.9935 ↑ 0.0001 |
| TissueMNIST | 0.9159 ↓ 0.0046 | 0.9058 ↑ 0.0009 | 0.8853 ↑ 0.0074 | 0.8495 ↑ 0.0472 | 0.8245 ↑ 0.0463 | 0.8135 ↑ 0.0432 |
| OrganAMNIST | 0.9968 ↑ 0.0003 | 0.9955 ↑ 0.0001 | 0.9952 ↑ 0.0007 | 0.9897 ↑ 0.0039 | 0.9925 ↑ 0.0007 | 0.9897 ↑ 0.0026 |
| OrganCMNIST | 0.9928 ↑ 0.0004 | 0.9865 ↑ 0.0025 | 0.9858 ↑ 0.0013 | 0.9829 ↑ 0.0032 | 0.9806 ↑ 0.0010 | 0.9805 ↑ 0.0010 |
| OrganSMNIST | 0.9770 ↑ 0.0004 | 0.9732 ↓ 0.0002 | 0.9664 ↑ 0.0038 | 0.9583 ↑ 0.0097 | 0.9595 ↑ 0.0026 | 0.9536 ↑ 0.0083 |

Table 15: AUROC on ten 2D datasets from MedMNISTv2 at different levels of uniform noise.

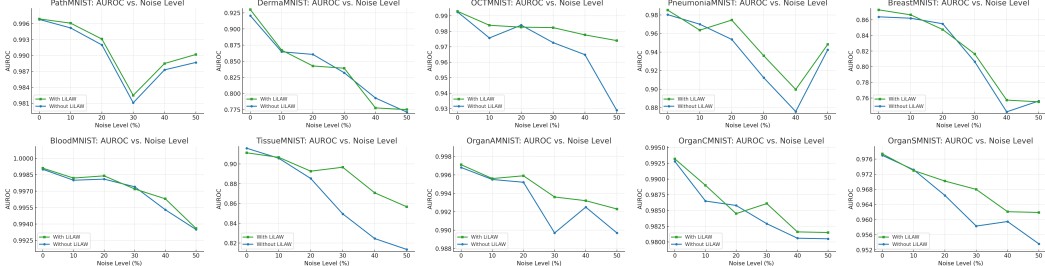

Figure 7: AUROC with and without LiLAW on ten 2D datasets from MedMNISTv2.

## A.16 COMPARISON TO BASELINES FOR IDENTIFYING MISLABELS

In Table 16, we evaluate mislabel detection on CIFAR-100 with 50% uniform noise, comparing LiLAW's weights to the mislabel flags (0 for mislabeled, 1 for correctly labeled) at an early training stage (epoch 3) and later training stage (epoch 10). We note that early-stage signals are very informative for LiLAW with $\mathcal{W}_\alpha$ achieving the best AUROC ($0.9838 \rightarrow 0.9782$) and the best AUPRC ($0.9810 \rightarrow 0.9755$) at both stages, with minor changes, compared to all 7 other methods.

This indicates that, early in training, confidently learned examples help expose mislabeled points (which remain high-loss), and $\mathcal{W}_\alpha$ captures this separation robustly over time. $\mathcal{W}_\beta$ attains high AUROC $(0.9719 \rightarrow 0.9768)$ but low AUPRC $(\approx 0.31$ at both epochs), suggesting it ranks many mislabeled items highly but also hard and correct samples highly. $\mathcal{W}_\delta$ improves AUROC with training $(0.8434 \rightarrow 0.9258)$ while maintaining low AUPRC $(\approx 0.33)$, suggesting that hard samples get easier to identify over time but there may also be an increase in false positives. Note that $\mathcal{W}$ takes the sum of $\mathcal{W}_\alpha$, $\mathcal{W}_\beta$, and $\mathcal{W}_\delta$ to aggregate signal from easy, moderate, and hard samples.

| Method | AUROC | | AUPRC | |
|---|---|---|---|---|
| | Epoch 3 | Epoch 10 | Epoch 3 | Epoch 10 |
| Data-IQ (Seedat et al., 2022) | 0.9363 | 0.8348 | 0.9290 | 0.9746 |
| DataMaps (Swayamdipta et al., 2020) | 0.5000 | 0.6989 | 0.5000 | 0.8320 |
| CNLCU-S (Xia et al., 2021) | 0.9438 | 0.9026 | 0.3119 | 0.3181 |
| AUM (Pleiss et al., 2020) | 0.9680 | 0.9635 | 0.9649 | 0.9610 |
| EL2N (Paul et al., 2023) | 0.9180 | 0.7567 | 0.3161 | 0.3604 |
| GraNd (Paul et al., 2023) | 0.6402 | 0.7034 | 0.4820 | 0.4054 |
| Forgetting (Toneva et al., 2019) | 0.5000 | 0.5782 | 0.5000 | 0.5740 |
| $\mathcal{W}_\alpha$ | **0.9838** | **0.9782** | **0.9810** | **0.9755** |
| $\mathcal{W}_\beta$ | 0.9719 | 0.9768 | 0.3086 | 0.3085 |
| $\mathcal{W}_\delta$ | 0.8434 | 0.9258 | 0.3454 | 0.3164 |
| $\mathcal{W}$ | 0.7103 | 0.9069 | 0.4985 | 0.3268 |

Table 16: AUROC and AUPRC comparison of metrics from our method ($\mathcal{W}_\alpha$, $\mathcal{W}_\beta$, $\mathcal{W}_\delta$, $\mathcal{W}$) to identify mislabels (in CIFAR-100 with 50% uniform noise) in early-stage training (epoch 3) and late-stage training (epoch 10) with 7 other difficulty estimation methods (Note: non-LiLAW metrics were obtained while training without LiLAW).

