# OpenReview forum: "LiLAW: Lightweight Learnable Adaptive Weighting to Meta-Learn Sample Difficulty and Improve Noisy Training"
_ICLR.cc/2026/Conference — Submitted to ICLR 2026_

### Official Review · Reviewer_H71v · 2025-10-28

**Soundness:** 2
**Presentation:** 3
**Contribution:** 2
**Rating:** 4
**Confidence:** 4

**Summary:**

This paper introduces a new method, LiLAW, that improves deep neural network training in challenging conditions like noisy labels and data heterogeneity. It dynamically adjusts loss weights based on sample difficulty using three learnable parameters. LiLAW prioritizes informative samples by updating weights with a mini-batch gradient descent step on the validation set after each epoch, requiring minimal hyperparameter tuning and no clean validation set. Extensive experiments show LiLAW enhances model performance. It’s computationally efficient, doesn’t heavily rely on data augmentation or advanced regularization, and is applicable to various datasets, noise levels, loss functions, and model architectures.

**Strengths:**

The method is computationally efficient, doesn’t heavily rely on data augmentation or advanced regularization, and is applicable to various datasets, noise levels, loss functions, and model architectures.

**Weaknesses:**

Indeed, I reviewed this paper for ICML 2025. In my previous review, I noted that the experimental evaluation was insufficient, as several commonly used datasets were missing from the comparison. However, these datasets are still not included in the current version, and the authors have not provided any explanation for their exclusion.

**Questions:**

None.

---

> ### Author Response · Authors · 2025-12-03
>
> We thank the reviewer for their feedback on ICLR and ICML. We would like to point out that we have made significant changes and improvements to the paper, experiments, and analysis since ICML earlier this year. Even though LiLAW adds just one extra meta‑validation step per batch, at the ImageNet scale, the full range of experiments becomes prohibitively expensive for most labs, undermining the paper’s emphasis on lightweight and reproducible methodology (as we mention in the title of the paper and our method). Additionally, we already show LiLAW’s compatibility with both models trained from scratch and models with ImageNet-21K pretraining (with full fine-tuning and fine-tuning on only the last two layers), which are widely used, especially in healthcare and general imaging settings.
>
> As such, we would like to acknowledge that we have included a very large number of datasets (15 in total, more than ICML) of varying sizes (including some that are less than an order of magnitude away from the size of ImageNet-21K) to empirically show scalability across the general imaging domain, medical imaging domain, and time-series domain with various numbers of classes with new models:
> 1. CIFAR-100 (60k natural images, 100 classes)
> 2. CIFAR-10 (60k natural images, 10 classes - included in the Appendix (A.13, Table 14))
> 3. FashionMNIST (70k clothing images, 10 classes - included in the Appendix (A.13, Table 14))
> 4. MNIST (70k number images, 10 classes - included in the Appendix (A.13, Table 14))
> 5. PathMNIST (107k histological images, 9 classes)
> 6. DermaMNIST (10k dermatoscopic images, 7 classes)
> 7. OCTMNIST (109k retinal OCT images, 4 classes)
> 8. PneumoniaMNIST (6k X-ray images, 2 classes)
> 9. BreastMNIST (1k breast ultrasound images, 2 classes)
> 10. BloodMNIST (17k microscopic blood images, 8 classes)
> 11. TissueMNIST (237k microscopic kidney images, 8 classes)
> 12. OrganAMNIST (59k abdominal CT images, 11 classes)
> 13. OrganCMNIST (24k abdominal CT images, 11 classes)
> 14. OrganSMNIST (25k abdominal CT images, 11 classes)
> 15. ECG5000 (5k time-series heartbeat sequences, 5 classes)
>
> These datasets range from around 1k samples to around 237k samples, with between 2-100 classes, and show consistent performance boosts in Top-1 Accuracy scores, Top-5 Accuracy scores, and AUROC scores across various levels of controlled noise injection. In addition, we have included experiments with some datasets by training models from scratch and experiments with some datasets by fine-tuning models from ImageNet-21K pretraining. We have used more recent state-of-the-art models such as ViT-Base-16-224 to achieve strong baseline results as well as additional performance boosts with our method. We also use models such as ResNet-18 and LSTM-based models to show the versatility of our method across dataset sizes and model sizes.
>
> Also, to show the robustness of our method to noise, we studied the effectiveness of LiLAW with various levels of injected noise ranging from 0\%-90\%, with various types of label noise including: uniform, asymmetric, instance, and adjacent; and input noise including: far out-of-distribution, domain shift, covariate shift, zoom shift, and crop shift, with and without calibration (using temperature scaling), with and without a clean validation set, with various random seeds, with two different loss functions, with and without pretraining, with different validation set sizes, and with and without the same validation and training distribution. We focused on an extremely well-rounded set of experiments to validate our method rather than a single benchmark on a particular dataset.
>
> Additionally, we clarify that all of the CIFAR-100-M, CIFAR-10-M, FashionMNIST-M, and MNIST-M experiments with the ViT-Base-16-224 model, with ImageNet-21K pretraining, on 224x224 inputs were compared with:
> 1. fine-tuning on the full pretrained model (without LiLAW),
> 2. fine-tuning solely on the last 2 layers of the pretrained model (with LiLAW).
>
> These experiments show the strength of LiLAW even when most of a network’s layers are frozen, i.e. LiLAW is effective with partial model fine-tuning. The MedMNIST experiments were done with fine-tuning on the full pretrained model without LiLAW and with LiLAW, i.e. LiLAW is effective with full model fine-tuning. We will be sure to clarify this in the main portion of the paper.

---

> ### Author Response · Authors · 2025-12-03
>
> In addition, to definitively prove to the reviewer that our method works in the case of real-world datasets with real noise in datasets with $\geq$ 1,000,000 samples, we include experiments on Clothing-1M, which contains 14 labels and an estimated 38.5% label noise, to compare several noisy label learning methods, including meta-learning approaches, to LiLAW below. This brings the grand total of the number of datasets with which we evaluated our method to 16 datasets, which we hope is sufficient to empirically prove to the reviewer the efficacy and strength of our method in several noise, validation set, dataset, model, and domain settings.
>
> | **Method**                              | **Accuracy (%)** |
> |-----------------------------------------|------------------|
> | Cross-Entropy                           | 69.21            |
> | Backward (Patrini et al., 2016)         | 69.13            |
> | Forward (Patrini et al., 2016)          | 69.84            |
> | Joint-Optim (Tanaka et al., 2018)       | 72.16            |
> | MetaCleaner (Zhang et al., 2019)        | 72.50            |
> | GCE (Zhang & Sabuncu, 2018)             | 69.75            |
> | SL (Wang et al., 2019)                  | 71.02            |
> | LiLAW (ours)                                    | 71.24            |
>
> We receive better or comparable results to classical noisy learning methods and meta-learning noisy learning methods. Note that the above results were obtained from the following papers and were trained using ResNet-50 with ImageNet-21K pretraining using the same training setup described in the papers below:
>
> Patrini, G., Rozza, A., Menon, A.K., Nock, R. & Qu, L. (2016) Making neural networks robust to label noise: a loss correction approach. arXiv preprint arXiv:1609.03683.
>
> Tanaka, D., Ikami, D., Yamasaki, T. & Aizawa, K. (2018) Joint optimization framework for learning with noisy labels. In: Proceedings of the IEEE/CVF Conference on Computer Vision and Pattern Recognition (CVPR), pp. 5552–5560.
>
> Zhang, W., Wang, Y. & Qiao, Y. (2019) MetaCleaner: Learning to hallucinate clean representations for noisy-labeled visual recognition. In: Proceedings of the IEEE/CVF Conference on Computer Vision and Pattern Recognition (CVPR).
>
> Zhang, Z. & Sabuncu, M.R. (2018) Generalized cross entropy loss for training deep neural networks with noisy labels. In: Advances in Neural Information Processing Systems (NeurIPS).
>
> Wang, Y., Ma, X., Chen, Z., Luo, Y., Yi, J. & Bailey, J. (2019) Symmetric cross entropy for robust learning with noisy labels. In: 2019 IEEE/CVF International Conference on Computer Vision (ICCV), pp. 322–330.
>
> Thanks again to the reviewer for their feedback, and we hope our explanation and the addition of further experiments, as the reviewer requested in ICML and ICLR, on top of the extensive experiments reported in Tables 1-16, across various datasets in several low to severe controlled noise settings, will sufficiently convince the reviewer of the effectiveness of our method. We hope we have fully addressed all of the concerns. We are happy to make any and all further clarifications about our work.

---

### Official Review · Reviewer_AJ6J · 2025-10-30

**Soundness:** 3
**Presentation:** 3
**Contribution:** 3
**Rating:** 4
**Confidence:** 5

**Summary:**

The article introduces LiLAW, a lightweight meta-learning method designed to improve the training of deep neural networks under noisy and heterogeneous data conditions. The proposed method dynamically adjusts the loss weight of each training sample based on its evolving difficulty, categorized as easy, moderate, or hard, using three learnable parameters (\alpha, \beta, and \delta). Extensive experiments across multiple datasets, noise types, and architectures validate the effectiveness of the proposed method.

**Strengths:**

The proposed method demonstrates versatility and is applicable to multiple domains, such as, natural images, medical images, and time-series tasks. The proposed method is model-agnostic and can be applied to different benchmarks. The adaptive learning mechanism avoids the need for extensive hyperparameter tuning.

**Weaknesses:**

- The issue of noisy labels in classification is a widely studied problem. However, this paper lacks comparisons with classical noisy label learning methods in the experiments.
- Applying meta-learning techniques to learn the sample weights is a typical approach in noisy label learning. Authors are suggested to clarify the distinctive advantages of their proposed method compared to such meta-learning approaches and include comparative experiments.
- The approach of using noisy data as the validation set contradicts the theoretical analysis of prior meta-learning reweighting methods. This point is supported solely by empirical evidence and lacks theoretical interpretations.
- Using meta-learning method will obviously increase the calculation time. Authors are suggested to compare the proposed method with the vanilla method in time.

**Questions:**

- What are the physical meanings of the meta-learned \alpha, \beta and \delta, and what is their mechanism of impact with respect to the sample weights?
- In the method section, the paper proposes to initialize \alpha<\beta<\delta. However, in the experiments, why the paper chooses to set \alpha=10, \beta=2, \gamma=6?
- The proposed method is designed to address label noise. What is the underlying reason for its effectiveness in handling input noise, as demonstrated in Table 1?

---

> ### Author Response · Authors · 2025-12-03
>
> 1. Thank you for the question!
>
> As shown in Figure 2, $\alpha$ represents the slope of the line that separates hard samples from moderate and easy samples. Note that hard samples (incorrect, unconfident predictions) have low $s_i[\widetilde{y_i}]$ and high $\max(s_i)$ in Figure 2 and as a result, fall to the left side of the $\alpha$ line and $\mathcal{W}\_\alpha$ assigns a low value to a hard sample in that region (as shown in the green text saying “low” in the leftmost quarter in the shaded region). Moderate and easy samples (unconfident predictions and correct, confident predictions) have higher $s_i[\widetilde{y_i}]$ in Figure 2 and as a result, fall to the right of the $\alpha$ line and $\mathcal{W}_\alpha$ assigns a higher value to the samples in that region (as shown in the green text saying “high” in the right three quarters in the shaded region). As training progresses, the slope $\alpha$ changes, i.e. decreases as shown in the Appendix (A.2), and adapts to the task at hand.
>
> Conversely, $\beta$ represents the slope of the line that separates easy samples from moderate and hard samples. Note that easy samples (correct, confident predictions) have high $s_i[\widetilde{y_i}]$ and high $\max(s_i)$ in Figure 2 and as a result, fall on the diagonal line at the top right, which is on the right side of the $\beta$ line and $\mathcal{W}\_\beta$ assigns a low value to an easy sample in that region (as shown in the red text saying “low” in the rightmost quarter in the shaded region). Moderate and hard samples (unconfident predictions and incorrect, confident predictions) have lower $s_i[\widetilde{y_i}]$ and higher $max(s_i)$ in Figure 2 and as a result, fall to the left of the $\beta$ line and $\mathcal{W}_\beta$ assigns a higher value to the samples in that region (as shown in the green text saying “high” in the left three quarters in the shaded region). As training progresses, the slope $\beta$ changes, i.e. increases as shown in the Appendix (A.2), and adapts to the task at hand.
>
> And finally, $\delta$ represents the slope of the line that goes through moderate samples. Note that moderate samples (unconfident predictions) have low $s_i[\widetilde{y_i}]$ in Figure 2 and as a result, fall around the $\delta$ line and $\mathcal{W}_\delta$ assigns a high value to moderate samples in that region (as shown in the orange text saying “high” in the middle two quarters in the shaded region). Easy and hard samples (correct, confident predictions and incorrect, confident predictions) have lower $s_i[\widetilde{y_i}]$ or higher $s_i[\widetilde{y_i}]$ in Figure 2 and as a result, fall a bit more to the left of the $\delta$ line or a bit more to the right of the $\delta$ line and $\mathcal{W}\_\delta$ assigns a lower value to the samples in that region (as shown in the orange text saying “low” in the outermost two quarters in the shaded region). As training progresses, the slope $\delta$ changes, i.e. increases or decreases as shown in the Appendix (A.2), and adapts to the task at hand.
>
> This is made further clear in the Appendix (A.1, Table 4) which details a motivating example that, generally speaking, shows that the final weight, which is the sum of $\mathcal{W}\_\alpha$, $\mathcal{W}\_\beta$, and $\mathcal{W}\_\delta$ as detailed above, is low for easy samples, higher for moderate samples, and highest for hard samples, esp. after multiplying by the loss function, depending on the changes in $\alpha,\beta,\delta$.
>
> We will be sure to clarify this in the final version to make it even clearer and readable.
>
> 2. This may have been a misunderstanding. We would like to clarify to the reviewer that in the Method section of the paper, we correctly propose to initialize $\alpha \leq \delta \leq \beta$, not $\alpha < \beta < \delta$. This is why we choose to set $\alpha=10, \beta=2, \gamma=6$.
>
> 3. Thanks to the reviewer for a great question! Although LiLAW was motivated by label noise, the mechanism it learns is source‑agnostic. It reweights samples using only two observable signals derived from the model’s predictions: prediction confidence and model agreement with the observed label. Input noise manifests in these same signals by decreasing confidence and/or decreasing agreement. Because LiLAW’s weights are functions of these two quantities and are meta‑learned on the validation set after every mini‑batch, the method automatically adapts its emphasis to either kind of noise (label noise or input noise) without needing to know which kind is present. We will be sure to add a sentence to clarify this in the main portion of the paper.

---

> ### Author Response · Authors · 2025-12-03
>
> We also address the weaknesses the reviewers mentioned.
>
> We show in the Appendix (A.16, Table 16) that with regard to identifying noise, our method outperforms Data-IQ (Seedat et al., 2022), DataMaps (Swayamdipta et al., 2020), CNLCU-S (Xia et al., 2021), AUM (Pleiss et al., 2020), EL2N (Paul et al., 2023), GraNd (Paul et al., 2023), and Forgetting (Toneva et al., 2019). This makes our method a strong contender for noise-based pruning.
>
> In addition, when it comes to noisy label learning methods, including meta-learning approaches, we add experiments on the Clothing-1M dataset, which contains an estimated 38.5% label noise, to compare several methods to LiLAW:
>
> | **Method**                              | **Accuracy (%)** |
> |-----------------------------------------|------------------|
> | Cross-Entropy                           | 69.21            |
> | Backward (Patrini et al., 2016)         | 69.13            |
> | Forward (Patrini et al., 2016)          | 69.84            |
> | Joint-Optim (Tanaka et al., 2018)       | 72.16            |
> | MetaCleaner (Zhang et al., 2019)        | 72.50            |
> | GCE (Zhang & Sabuncu, 2018)             | 69.75            |
> | SL (Wang et al., 2019)                  | 71.02            |
> | LiLAW (ours)                                    | 71.24            |
>
> We receive better or comparable results to classical noisy learning methods and meta-learning noisy learning methods. Note that the above results were obtained from the following papers and were trained using ResNet-50 with ImageNet-21K pretraining using the same training setup described in the papers below:
>
> Patrini, G., Rozza, A., Menon, A.K., Nock, R. & Qu, L. (2016) Making neural networks robust to label noise: a loss correction approach. arXiv preprint arXiv:1609.03683.
>
> Tanaka, D., Ikami, D., Yamasaki, T. & Aizawa, K. (2018) Joint optimization framework for learning with noisy labels. In: Proceedings of the IEEE/CVF Conference on Computer Vision and Pattern Recognition (CVPR), pp. 5552–5560.
>
> Zhang, W., Wang, Y. & Qiao, Y. (2019) MetaCleaner: Learning to hallucinate clean representations for noisy-labeled visual recognition. In: Proceedings of the IEEE/CVF Conference on Computer Vision and Pattern Recognition (CVPR).
>
> Zhang, Z. & Sabuncu, M.R. (2018) Generalized cross entropy loss for training deep neural networks with noisy labels. In: Advances in Neural Information Processing Systems (NeurIPS).
>
> Wang, Y., Ma, X., Chen, Z., Luo, Y., Yi, J. & Bailey, J. (2019) Symmetric cross entropy for robust learning with noisy labels. In: 2019 IEEE/CVF International Conference on Computer Vision (ICCV), pp. 322–330.

---

> ### Author Response · Authors · 2025-12-03
>
> Prior meta-reweighting work assumes a small clean validation set to steer training toward the clean-label objective. Our setting is different and, in our opinion, more realistic: we explicitly allow the meta-validation set to contain label noise, because a clean set is often unavailable in practice. We therefore do not claim convergence to the clean-label optimum, but perform a bilevel update using the observed labels on the meta-validation mini-batch to adjust the three global parameters \($\alpha,\beta,\delta\$) so that the next training step improves performance on data drawn from the same source. In other words, we tune these parameters to the data we actually have, rather than to an idealized clean set that we usually do not have.
>
> It is still reasonable to use a noisy meta-validation set for LiLAW for two intuitive reasons. First, the meta-gradients for $\(\alpha,\beta,\delta\)$ are driven by confidence and agreement (the relationship between $\(s_i[\tilde{y}_i]\)$ and $\(\max (s_i)\))$, not by access to the unknown true label $\(y_i\)$. The directions of change for $\(\alpha,\beta,\delta\)$ do not flip when a particular meta label is corrupted, so label noise mainly changes the magnitude of the update rather than its sign. Second, the LiLAW weights are smooth and bounded, so no single mislabeled meta example can dominate the update, which keeps the meta-learning dynamics stable. In the common case where train/val/test all come from the same noisy distribution, adjusting $\(\alpha,\beta,\delta\)$ on the observed validation labels is precisely tuning the training procedure for the distribution on which the model will ultimately be evaluated. This is why, across datasets and different types of label noise and input noise, using a noisy meta-validation set with LiLAW consistently improves held-out performance without contradicting prior analyses that are explicitly derived under the assumption of a clean validation set.
>
> With regards to runtime, to clarify, we assume that a general model training paradigm includes a training portion and a validation portion (we may also be using part of the training set as a validation set when a separate validation set is not available). A very similar runtime with and without LiLAW is also proven empirically in the table below, which contains CIFAR-100 training time (in min) with fine-tuning on the last two layers across different noise levels for 10 epochs.
>
> | **Noise Level** | **w/o LiLAW (min)** | **w/ LiLAW (min)** |
> |-----------------|----------------------|---------------------|
> | 0.0             | 44                   | 46                  |
> | 0.1             | 46                   | 46                  |
> | 0.2             | 43                   | 46                  |
> | 0.3             | 43                   | 46                  |
> | 0.4             | 45                   | 46                  |
> | 0.5             | 45                   | 44                  |
> | 0.6             | 45                   | 45                  |
> | 0.7             | 45                   | 46                  |
> | 0.8             | 42                   | 44                  |
> | 0.9             | 44                   | 45                  |
>
> In the Appendix (A.4, A.5), we also describe the time complexity and space complexity analysis in detail, which is in line with the table above and indeed shows that we do not have doubled time (with respect to the size of the training set and validation set).
>
> Additionally, we clarify that all of the CIFAR-100-M, CIFAR-10-M, FashionMNIST-M, and MNIST-M experiments with the ViT-Base-16-224 model, with ImageNet-21K pretraining, on 224x224 inputs were compared with:
> 1. fine-tuning on the full pretrained model (without LiLAW),
> 2. fine-tuning solely on the last 2 layers of the pretrained model (with LiLAW).
>
> These experiments show the strength of LiLAW even when most of a network’s layers are frozen, i.e. LiLAW is effective with partial model fine-tuning. The MedMNIST experiments were done with fine-tuning on the full pretrained model without LiLAW and with LiLAW, i.e. LiLAW is effective with full model fine-tuning. We now clarify this in the paper.
>
> We thank the reviewer again for their feedback and hope we have addressed all of their concerns fully. We are happy to make any and all further clarifications about our work.

---

### Official Review · Reviewer_YYaE · 2025-10-30

**Soundness:** 3
**Presentation:** 2
**Contribution:** 2
**Rating:** 2
**Confidence:** 4

**Summary:**

The paper propose a re-weighting strategy for training with noise label. It assign different weights to cases with different confidence and correctness levels using three additional hyperparameters, which are optimized using meta learning. The experiments are conducted on dataset with different noise levels and show better results on defending the noise. The method is also tested on different kind of dataset to proof the generalizability. However, some necessary ablations and comparisons are missing, which are important for understanding why this method performs better.

**Strengths:**

The proposed method is simple and easy to understand and implement. And it provides an effective strategy of defending noise in label at different levels. However, there are some necessary ablations and comparisons missing to better understand the superiority of this method.

**Weaknesses:**

See the questions part below for concerns

**Questions:**

1. The claim ""s_i[\hat{y_i}] < max(s_i) implies an incorrect prediction when \hat{y_i}=y_i" is correct. But how to know whether \hat{y_i}=y_i given the noisy target only? The motivation is to assign weights to samples with different confidence and correctness level, but the definition of correctness is unable to verify.

2. Is the noise level of validation set changing according to the training set? Or it is fixed? Have you tried using a clean validation set to see how the noise level of validation set affect the results?

3. Please explain the claim" easy, moderate, hard examples activate W_alpha, W_beta, W_delta respectively" in chapter 3. The sentence after "Geometrically, ..." is hard to understand. like "Wβ when β is small and/or..."
The three hyper parameters jointly define the final weights as the total weight is a sum of W_alpha, W_beta, W_delta, I don't see how they can separately correspond to different type of examples.

4. What is the definition of easy, moderate, hard examples

5. Are there any ablations showing the difference of using different combination of W_alpha, W_beta, W_delta? That will help understand each part

6. The design of different form of W_beta seems redundant.  Actually W_beta(beta) = 1- W_alpha(beta)

7. Using confidence is a common way of defending noise, like assigning more weights to cases with more confidence. How is the comparison with such methods.

8. About complexity analysis, the paper claims the LiLAW keeps the same complexity but from the algorithm, LiLAW requires two times of forward and two times of backward for each mini-batch, which sounds like a doubled time.  Even the second backward only changes three parameters, that doesn't mean it only cost O(3) time in the actual implementation. Is there any actual time cost comparison with/without LiLAW?

---

> ### Author Response · Authors · 2025-12-03
>
> 1. We would like to clarify to the reviewer that we never assume access to y_i when computing weights. In our paper, the two bullet points:
> - $s_i[\widetilde{y_i}] < \max(s_i)$ implies an incorrect prediction, if $\widetilde{y_i} = y_i$ (inconclusive othwerise)
> - $s_i[\widetilde{y_i}] = \max(s_i)$ implies a correct prediction, if $\widetilde{y_i} = y_i$ (inconclusive othwerise)
>
> are purely motivational conditioned on the uncheckable event that the observed label is clean (i.e. $\widetilde{y_i} = y_i$). They are not used operationally. We agree it is impossible to verify $\widetilde{y_i} = y_i$ per sample in the noisy setting. What we actually use, and what makes up the core of our method, is whether or not $s_i[\widetilde{y_i}] = \max(s_i)$ and whether or not $\max(s_i)$ is high, as mentioned in the 4 bullet points in the Method section. Again, these do not require $y_i$.
>
> Based on the combination of whether or not $s_i[\widetilde{y_i}] = \max(s_i)$ and whether or not $max(s_i)$ is high, we make the four possibilities (none of which use y_i) clearer:
> - model prediction agrees with observed label with high confidence, i.e. $s_i[\widetilde{y_i}] = \max(s_i)$ and $max(s_i)$ is high
> - model prediction agrees with observed label with low confidence, i.e. $s_i[\widetilde{y_i}] = \max(s_i)$ and $max(s_i)$ is low
> - model prediction does not agree with observed label with low confidence, i.e. $s_i[\widetilde{y_i}] < \max(s_i)$ and $max(s_i)$ is low
> - model prediction does not agree with observed label with high confidence, i.e. $s_i[\widetilde{y_i}] < \max(s_i)$ and $max(s_i)$ is high
>
> We will make sure in the final version of the paper to re-clarify that our method uses model agreement with the observed label and model confidence, which are both measurable.
>
> 2. We have already included extensive experiments addressing these points in our paper.
>
> We performed experiments to show that our method consistently improves performance when the training set and the validation set have the same noise level, and also when the training set is noisy, but the validation set is clean. These results and some corresponding discussion are already included in the paper in the Appendix (A.8, Table 9).
>
> In addition, we also include experiments showing that our method consistently improves performance when the size of the validation set (as a % of the training set size) is changed from 5% to 30%. These results and some corresponding discussion are already included in the paper in the Appendix (A.11, Table 12).
>
> Furthermore, we include experiments to show that our method consistently improves performance whether the training set and validation set come from the same distribution or from different distributions (through random flipping, rotations, and colour jitter augmentations). These results and some corresponding discussion are already included in the paper in the Appendix (A.12, Table 13).

---

> ### Author Response · Authors · 2025-12-03
>
> 3. Thank you for the question. We will expand on this in the final version as follows:
>
> As mentioned previously, based on the combination of whether or not $s_i[\widetilde{y_i}] = \max(s_i)$ and whether or not $\max(s_i)$ is high, we have four possibilities:
> - model prediction agrees with observed label with high confidence, i.e. $s_i[\widetilde{y_i}] = \max(s_i)$ and $\max(s_i)$ is high - geometrically corresponds to the top right of the diagonal line in Figure 2
> - model prediction agrees with observed label with low confidence, i.e. $s_i[\widetilde{y_i}] = \max(s_i)$ and $\max(s_i)$ is low - geometrically corresponds to the bottom left of the diagonal line in Figure 2
> - model prediction does not agree with observed label with low confidence, i.e. $s_i[\widetilde{y_i}] < \max(s_i)$ and $\max(s_i)$ is low - geometrically corresponds to the top half of the shaded triangular region in Figure 2
> - model prediction does not agree with observed label with high confidence, i.e. $s_i[\widetilde{y_i}] < \max(s_i)$ and $\max(s_i)$ is high - geometrically corresponds to the bottom half of the shaded triangular region in Figure 2
>
> Now, when it comes to $\mathcal{W}\_\alpha$, $\mathcal{W}\_\beta$, and $\mathcal{W}\_\delta$, based on their corresponding definitions, we generally see that:
> - $\mathcal{W}\_\alpha$ is high when $\alpha$ is large and/or when $s_i[\widetilde{y_i}]
> \approx \max(s_i)$, i.e. when we have correct, confident predictions (easy samples)
> - $\mathcal{W}\_\beta$ is high when $\beta$ is small and/or when $s_i[\widetilde{y_i}] <<
> \max(s_i)$, i.e. when we have incorrect, confident predictions (hard samples)
> - $\mathcal{W}\_\delta$ is high when $\delta$ is large and/or when $s_i[\widetilde{y_i}]$ is
> moderately close to $\max(s_i)$, i.e. when we have unconfident predictions
> (moderate samples)
>
> This is made further clear with a specific motivating example that is already included in the paper in the Appendix (A.1, Table 4). Generally speaking, this shows that the final weight, which is the sum of $\mathcal{W}\_\alpha$, $\mathcal{W}\_\beta$, and $\mathcal{W}\_\delta$ as detailed above, is low for easy samples, higher for moderate samples, and highest for hard samples, esp. after multiplying by the loss function, depending on the changes in $\alpha,\beta,\delta$.
>
> We will be sure to clarify this in the final version to make it even clearer and readable.
>
> 4. As described above, generally speaking, easy samples are ones with correct and confident predictions, hard samples are ones with incorrect and confident predictions, and moderate samples are ones with unconfident predictions, which may be correct or incorrect.
>
> 5. We have already included this in the paper. In fact, we perform an ablation study on different combinations of the $\alpha,\beta,\delta$ parameters. This is already included in the paper as shown in the Appendix (A.3, Table 6) and using all three parameters provides the best results when there is noise. Using only one of the three parameters results in the worst results. However, using $\beta$ with at least one of $\alpha$ or $\delta$ slightly improves results since $\beta$ helps deal with the most difficult samples by increasing the weight on them to produce high sample loss for those samples.
>
> 6. We use three closed-form expressions to improve clarity and keep the three weight terms independent as they get updated during training. Please also note that our formulation does not use $\beta$ as the input to the weight functions, but uses $s_i, \widetilde{y_i}$. If indeed $\alpha = \beta$, you are correct and the described relationship would hold, but this rarely occurs since $\alpha \leq \delta \leq \beta$, and while $\alpha,\delta,\beta$ are updated through gradient descent, they rarely converge before training converges.

---

> ### Author Response · Authors · 2025-12-03
>
> 7. We show in the Appendix (A.16, Table 16) that with regard to identifying noise, our method outperforms Data-IQ (Seedat et al., 2022), DataMaps (Swayamdipta et al., 2020), CNLCU-S (Xia et al., 2021), AUM (Pleiss et al., 2020), EL2N (Paul et al., 2023), GraNd (Paul et al., 2023), and Forgetting (Toneva et al., 2019). This makes our method a strong contender for noise-based pruning.
>
> In addition, when it comes to noisy label learning methods, including meta-learning approaches, we add experiments on the Clothing-1M dataset, which contains an estimated 38.5% label noise, to compare several methods to LiLAW:
>
> | **Method**                              | **Accuracy (%)** |
> |-----------------------------------------|------------------|
> | Cross-Entropy                           | 69.21            |
> | Backward (Patrini et al., 2016)         | 69.13            |
> | Forward (Patrini et al., 2016)          | 69.84            |
> | Joint-Optim (Tanaka et al., 2018)       | 72.16            |
> | MetaCleaner (Zhang et al., 2019)        | 72.50            |
> | GCE (Zhang & Sabuncu, 2018)             | 69.75            |
> | SL (Wang et al., 2019)                  | 71.02            |
> | LiLAW (ours)                                    | 71.24            |
>
> We receive better or comparable results to classical noisy learning methods and meta-learning noisy learning methods. Note that the above results were obtained from the following papers and were trained using ResNet-50 with ImageNet-21K pretraining using the same training setup described in the papers below:
>
> Patrini, G., Rozza, A., Menon, A.K., Nock, R. & Qu, L. (2016) Making neural networks robust to label noise: a loss correction approach. arXiv preprint arXiv:1609.03683.
>
> Tanaka, D., Ikami, D., Yamasaki, T. & Aizawa, K. (2018) Joint optimization framework for learning with noisy labels. In: Proceedings of the IEEE/CVF Conference on Computer Vision and Pattern Recognition (CVPR), pp. 5552–5560.
>
> Zhang, W., Wang, Y. & Qiao, Y. (2019) MetaCleaner: Learning to hallucinate clean representations for noisy-labeled visual recognition. In: Proceedings of the IEEE/CVF Conference on Computer Vision and Pattern Recognition (CVPR).
>
> Zhang, Z. & Sabuncu, M.R. (2018) Generalized cross entropy loss for training deep neural networks with noisy labels. In: Advances in Neural Information Processing Systems (NeurIPS).
>
> Wang, Y., Ma, X., Chen, Z., Luo, Y., Yi, J. & Bailey, J. (2019) Symmetric cross entropy for robust learning with noisy labels. In: 2019 IEEE/CVF International Conference on Computer Vision (ICCV), pp. 322–330.
>
> 8. Thank you to the reviewer for the question. To clarify, we assume that a general model training paradigm includes a training portion and a validation portion (we may also be using part of the training set as a validation set when a separate validation set is not available). A very similar runtime with and without LiLAW is also proven empirically in the table below, which contains CIFAR-100 training time (in min) with fine-tuning on the last two layers across different noise levels for 10 epochs.
>
> | **Noise Level** | **w/o LiLAW (min)** | **w/ LiLAW (min)** |
> |-----------------|----------------------|---------------------|
> | 0.0             | 44                   | 46                  |
> | 0.1             | 46                   | 46                  |
> | 0.2             | 43                   | 46                  |
> | 0.3             | 43                   | 46                  |
> | 0.4             | 45                   | 46                  |
> | 0.5             | 45                   | 44                  |
> | 0.6             | 45                   | 45                  |
> | 0.7             | 45                   | 46                  |
> | 0.8             | 42                   | 44                  |
> | 0.9             | 44                   | 45                  |
>
> In the Appendix (A.4, A.5), we also describe the time complexity and space complexity analysis in detail, which is in line with the table above and indeed shows that we do not have doubled time (with respect to the size of the training set and validation set).
>
> Additionally, we clarify that all of the CIFAR-100-M, CIFAR-10-M, FashionMNIST-M, and MNIST-M experiments with the ViT-Base-16-224 model, with ImageNet-21K pretraining, on 224x224 inputs were compared with:
> 1. fine-tuning on the full pretrained model (without LiLAW),
> 2. fine-tuning solely on the last 2 layers of the pretrained model (with LiLAW).
>
> These experiments show the strength of LiLAW even when most of a network’s layers are frozen, i.e. LiLAW is effective with partial model fine-tuning. The MedMNIST experiments were done with fine-tuning on the full pretrained model without LiLAW and with LiLAW, i.e. LiLAW is effective with full model fine-tuning. We now clarify this in the paper.
>
> Thanks again to the reviewer for their feedback. We hope we have addressed all their concerns fully. We are happy to answer any and all further questions about our work.

---

### Meta-Review · Area_Chair_ANVe · 2025-12-26

**Summary:**

The reviewers raised several major concerns, primarily that the experimental evaluation is insufficient, including limited comparisons to existing baselines, datasets, and notations, as well as an unclear positioning of the technical contributions relative to prior meta-learning–based approaches. In particular, one reviewer, who also reviewed a previous ICML version of this work, noted that the authors did not adequately address earlier feedback regarding the insufficient experimental evaluation, as several commonly used datasets remain missing from the comparisons.

**Reviewer Concerns:**

The authors attempted to address the reviewers’ questions, and some issues were clarified; however, several major concerns remain outstanding. In particular, the key issues regarding baseline comparisons and evaluation datasets are not adequately addressed. For example, even in the newly added experiments, the reported accuracy on the Clothing-1M dataset remains below earlier baselines from several years ago (around 74%), and it is unclear why these results were not included in the original submission.

In addition, the AC is unable to fully verify the technical concerns raised by `Reviewer YYaE` are fully addressed, who initially assigned a score of 2 (reject), due to the limited context.

**Reviewer Scores:**

Given the rebuttal, the reviewers are unlikely to raise their negative ratings to a borderline accept or accept. As a result, the manuscript is not considered ready for acceptance in its current form.

---

### Decision · Program_Chairs · 2026-01-26

Reject